# Unleashing the Potential of Multimodal LLMs for Zero-Shot Spatio-Temporal Video Grounding

**Zaiquan Yang**    **Yuhao Liu**[†]    **Gerhard Hancke**    **Rynson W.H. Lau**[†]

Department of Computer Science
City University of Hong Kong

{zaiquyang2-c, yuhliu9-c}@my.cityu.edu.hk
{gp.hancke, Rynson.Lau}@cityu.edu.hk

## Abstract

Spatio-temporal video grounding (STVG) aims at localizing the spatio-temporal tube of a video, as specified by the input text query. In this paper, we utilize multimodal large language models (MLLMs) to explore a zero-shot solution in STVG. We reveal two key insights about MLLMs: (1) MLLMs tend to dynamically assign special tokens, referred to as *grounding tokens*, for grounding the text query; and (2) MLLMs often suffer from suboptimal grounding due to the inability to fully integrate the cues in the text query (*e.g.*, attributes, actions) for inference. Based on these insights, we propose a MLLM-based zero-shot framework for STVG, which includes novel decomposed spatio-temporal highlighting (DSTH) and temporal-augmented assembling (TAS) strategies to unleash the reasoning ability of MLLMs. The DSTH strategy first decouples the original query into attribute and action sub-queries for inquiring the existence of the target both spatially and temporally. It then uses a novel logit-guided re-attention (LRA) module to learn latent variables as spatial and temporal prompts, by regularizing token predictions for each sub-query. These prompts highlight attribute and action cues, respectively, directing the model's attention to reliable spatial and temporal related visual regions. In addition, as the spatial grounding by the attribute sub-query should be temporally consistent, we introduce the TAS strategy to assemble the predictions using the original video frames and the temporal-augmented frames as inputs to help improve temporal consistency. We evaluate our method on various MLLMs, and show that it outperforms SOTA methods on three common STVG benchmarks. The code will be available at `https://github.com/zaiquanyang/LLaVA_Next_STVG`.

## 1 Introduction

Spatio-Temporal Video Grounding (STVG) aims to localize a target object in a video both spatially and temporally, given an input text query. This task is fundamental to many different applications (*e.g.*, video surveillance and autonomous driving [67]). However, it is also very challenging as it requires the model to be able to distinguish the target from distractors over time and identify the precise temporal boundary of the action. While existing methods handle the STVG task mainly in a fully-supervised setting [13; 59; 54], which often relies on costly frame-level annotations, several works [29; 3; 61; 21; 60; 25] attempt to introduce the weakly-supervised or zero-shot setting to alleviate the burden of dense annotations. For example, E3M [3] integrates CLIP [43] and an expectation maximization strategy to optimize spatio-temporal localization in a zero-shot manner. However, CLIP is known to be weak in localization [75; 14] as it simply aligns the global representation of image-text pairs.

---

[†] Joint corresponding authors.

39th Conference on Neural Information Processing Systems (NeurIPS 2025).

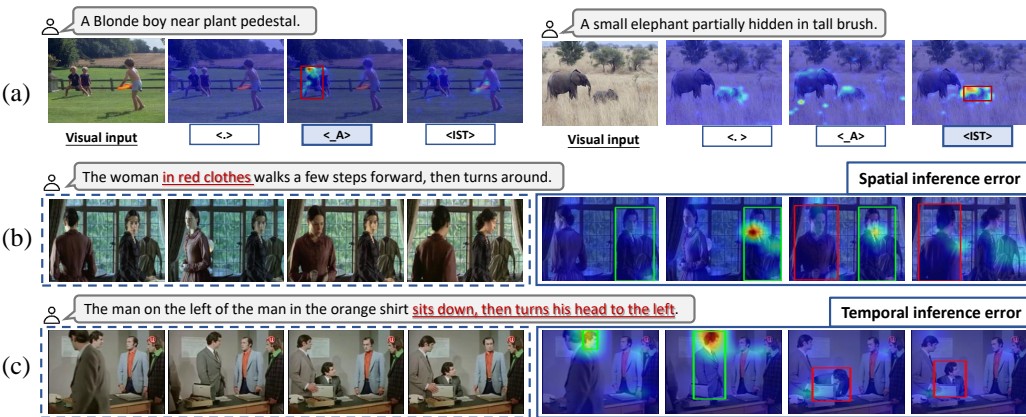

Figure 1: (a) The visual attention maps shows that some special tokens (marked as ▭) can precisely attend to the target region of the input query. However, these special tokens, referred to as the *grounding tokens* in our work, underperform on complex STVG, where they often focus on part cues and ignore other cues (marked in red within the input text prompts). Examples (b) and (c) illustrate spatial/temporal grounding errors caused by ignoring the discriminative attribute/action cues. Red and green bounding boxes denote the ground truths and predictions, respectively.

Considering the strong capability of multi-modal large language models (MLLMs) [37; 27; 31; 8; 53; 36] in cross-modality alignment, several works explore the application of MLLMs in the visual grounding task. However, they typically require explicit fine-tuning [26; 44; 39] of MLLMs with additional grounding datasets and specific model modifications, which can be difficult to scale and generalize to novel visual data [4]. Although some recent works [4; 69] have investigated the attention maps in language models as done in previous ViT and CNN architectures [74; 52; 12], they only focus on the generated tokens while neglecting other token components input in the language model (*e.g.*, system tokens and special tokens). In particular, we observe that the special tokens play an important role in structuring the communication between the user input and the language model, and help guide the generation of coherent responses in dialogues.

With the above observation in mind, we delve deeper and find that the special tokens following the input instruction have outstanding grounding ability. In particular, a few special tokens are characterized by high visual activation and can attend to the region of interest well. Considering the left sample of Fig. 1(a) as an example, the special token '`_A`' provides tangible attention to the *'boy'* referred to by the given query, while in the right sample of Fig. 1(a), the token '`IST`' is assigned to ground the *'elephant'*. These special tokens are referred to as **grounding tokens** in our work. Despite the strong comprehension ability, the grounding tokens cannot optimally adapt to STVG as they tend to ignore some important cues (*e.g.*, attribute or action) in a complex video query. As shown in Fig. 1(b), the grounding token fails in the spatial grounding by ignoring the attributes. Similarly, in Fig. 1(c), it leads to the failure of temporal grounding by neglecting the action cues of the target.

In this work, we first conduct a systematic analysis to probe the special tokens of various MLLMs (Sec. 3.2), which demonstrates our aforementioned empirical findings and enables zero-shot spatio-temporal video grounding. To alleviate the problem of neglecting discriminative cues, we propose a novel decomposed spatio-temporal highlighting (DSTH) strategy (Sec. 3.3). Specifically, the language query is decomposed into attribute and action sub-queries, which are utilized as the text input of the MLLM for inquiring the target's existence spatially and temporally, respectively. We then propose a novel logit-guided re-attention (LRA) module to highlight the cues in attribute and action sub-queries. For each sub-query, LRA optimizes the learnable latent variable as the (spatial/temporal) prompts via enhancing the positive response generation while suppressing the negative response. As a result, the DSTH strategy can well adapt the model to mine faithful visual context and concentrate on spatially/temporally relevant regions. In addition, to further enhance the temporal consistency during spatial grounding by the attribute sub-query, we develop a temporal-augmented assembling (TAS) strategy (Sec. 3.4). The TAS strategy utilizes temporally perturbed frames as input to assemble different predictions into final spatial grounding result.

In summary, our contributions are as follows:

- We reveal that MLLMs dynamically assign the special tokens for precisely grounding text-related regions. We identify the special tokens characterized by the salient visual activation for deriving a novel zero-shot STVG framework.

- We propose a novel test-time tuning strategy named decomposed spatio-temporal highlighting (DSTH). It introduces an innovative logit-guided re-attention (LRA) module to adapt the grounding token for thorough spatio-temporal localization. We also develop a temporal-augmented assembling (TAS) strategy to further improve the robustness of spatial localization.

- We conduct extensive experiments on various MLLMs to demonstrate our empirical findings and validate the effectiveness of the proposed method. Our method outperforms existing methods by a remarkable margin on three STVG benchmarks.

## 2  Related work

**Spatio-Temporal Video Grounding (STVG)** aims to localize a spatio-temporal tube in a video corresponding to the text query. Unlike image-based visual grounding [10; 11; 63; 34; 62; 35], STVG [73; 51; 59] presents significant challenges, requiring models to distinguish targets from distractors both spatially and temporally. Fully-supervised STVG approaches [51; 54; 73; 49] have achieved promising results. However, these methods heavily depend on an extensive collection of labor-intensive annotations. Several recent works [29; 21; 25] tackle STVG in a more efficient manner, which only uses coarse video-level descriptions for training. E3M [3] leverages pre-trained vision-language models and proposes an expectation maximization framework to optimize spatio-temporal localization. However, contrastive objective based multimodal models are known to be weak in localization [75; 14] as they simply align the global representations of image-text pairs.

**Multimodal Large Language Models (MLLMs)** are capable of handling diverse language and vision tasks. To equip MLLMs with the grounding ability, current works [41; 26; 28; 66; 71] construct grounding-oriented supervision data for instruction tuning and propose novel architectural modifications. LLaVA-ST [28] achieves spatio-temporal understanding by introducing a progressive training strategy consisting of three sequential stages. However, the large amount of grounding data needed for instruction-tuning imposes high labeling costs. In addition, changing the focus of MLLMs to grounding tasks can degrade the original dialog capabilities due to catastrophic forgetting [68]. Recent works [4; 19; 69] reveal the inherent perception ability of MLLMs obtained by general instruction-tuning. Unlike these works that only focus on the generated tokens or in our work, we delve deeper into the more token components and reveal that the MLLMs always dynamically assign the special tokens following the instruction prompt for attending to the regions of interest.

**Test-time Tuning (TTT)** aims to optimize the inference of test samples online. With the progress of multimodal foundation models [43; 5]), TTT has attracted more attention [76; 40; 18; 48; 64] as it can learn effective prompts for test samples and well adapt the foundation model for zero-shot applications. TPT [48] pioneers the study on TTT by minimizing the prediction entropy between each test sample and its augmented views. HisTPT [70] explores memory learning for test-time prompt tuning by introducing the memory bank. However, the test prompt optimization for MLLMs is barely explored. The most relevant work to our work is ControlMLLM [56], which optimizes the attention map by taking the referring regions as supervision. With the lack of supervision, we propose to learn visual prompts by regularizing the token-level response to instruction input. Our work shows that we can rectify where text prompts attend to by altering the outputs of MLLMs.

## 3  Method

### 3.1  Preliminaries

**Task Formulation.** Given an untrimmed video $V = \{f_t\}_{t=1}^{T_v}$ composed of $T_v$ image frames and a sentence query $Q$, the goal of the STVG task is to localize the spatio-temporal tube of target $O = \{b_t\}_{t=t_s}^{t_e}$ described by $Q$. Here $b_t$ represents the bounding box of the target in the $t$-th frame, $t_s$ and $t_e$ specify the starting and ending boundaries of the target tubelet, respectively. The STVG task can be solved through two sub-tasks: spatial grounding and temporal grounding. In our work, we first derive a zero-shot solution for the STVG task by unleashing the strong cross-modal comprehension ability of MLLMs.

**The Setup of MLLMs**. Current MLLMs typically consist of a visual encoder, a projector, and a large language model. Specifically, given an image-question pair $(I, Q)$ as input, the image $I$ is first projected into text-aligned visual tokens $T_v = \{v_1, \ldots, v_M\}$ by visual encoder and projector, and the question $Q$ is converted into text tokens $T_q = \{t_1, \ldots, t_{N_q}\}$ by a text tokenizer and embedding layer. Here, $M$ and $N_q$ denote the numbers of visual tokens and question tokens, respectively. In practice, MLLMs also introduce system tokens $T_{sys}$ and some special tokens $T_{role}$ for instruction-following ability. In particular, the special tokens play an important role in structuring the well-organized conversation framework. In our work, the special tokens are positioned subsequent to the instruction prompts by the user. They help guide the generation of coherent responses. The number of special tokens $N_{role}$ often differs among different MLLMs. As a result, the language model receives concatenated tokens, $T = \{t_1, \ldots, t_{N_{sys}}; v_1, \ldots, v_M; t_1, \ldots, t_{N_q}; t_1, \ldots, t_{N_{role}}\}$, as input. Appendix 6.1 provides visual illustration about the input tokens in MLLMs.

**Text-to-visual Attention.** The LLM in MLLMs typically processes the input tokens through L transformer blocks [37] with the multi-head attention (MHA) for interactions of different tokens. Particularly, the text-to-visual attention represents the relationships between the visual and the textual tokens. We can derive the text-to-visual attention matrix $A \in \mathbb{R}^{L \times H \times N \times N}$, where $N = N_{sys} + M + N_q + N_{role}$. L and H denote the numbers of layers and heads in the transformer. Our empirical observation from Fig. 1 is that the special tokens $T_{role}$ show outstanding grounding ability with a global comprehension. For the simplicity of the following analysis, we omit the affect of different layers and heads by the mean operation. We can then obtain the text-to-visual attention matrix, $A_{role} = [N_{sys} + M + N_q : N, N_{sys} : N_{sys} + M]$.

### 3.2 Grounding Token Identification

In this section, we conduct a pilot study to quantitatively analyze the grounding ability of the special tokens. Without loss of generality, we randomly select a subset of $1,000$ image-text pairs from the RefCOCOg [17] validation set for image MLLMs analysis, and a subset of $1,000$ video-text pairs from the HC-STVGv2 [51] dataset for video MLLMs analysis. Particularly, we choose three typical MLLMs (*i.e.*, LLaVA-1.5, Qwen-VL, Deepseek-VL) for the study on image input and three MLLMs (*i.e.*, LLaVA-Next-Video, Qwen2-VL, LLaVA-OneVision) for the study on video input. We have two key findings from our studies.

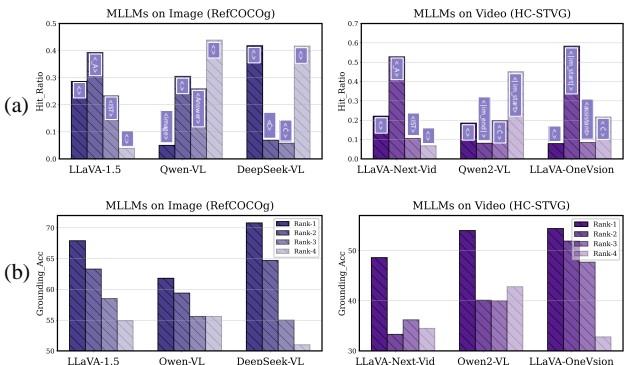

Figure 2: In (a), the results show the frequency with which different special tokens (such as '\_A', '.') have superior grounding ability than other tokens, *i.e.*, hit_ratio. In (b), the results represent the grounding accuracy of tokens ranked by their visual activation degrees. For each MLLM, we select four tokens for visualization.

**MLLMs dynamically assign the special tokens to attend to the text-related regions.** To demonstrate the finding, we first define the *attention ratio* of each special token as the ratio of maximum attention within the ground-truth bounding box $b_{gt}$ to that outside it. For example, the attention ratio of the special token '**\_A**' can be computed as:

$$R_{att}^{\text{-A}} = \frac{\max(A_{role}^{\text{-A}} \odot f_{B2M}(b_{gt}))}{\max\left(A_{role}^{\text{-A}} \odot (1 - f_{B2M}(b_{gt}))\right)}, \qquad A_{role}^{\text{-A}} \in \mathbb{R}^{1 \times M}, \tag{1}$$

where $f_{B2M}$ denotes the function that transforms a bounding box into its corresponding binary mask. Here, a higher ratio indicates a better target grounding ability. Given a test sample, we can identify the token yielding the highest attention ratio as the superior token for grounding. The ***hit ratio*** of a token is then defined as the frequency of being the superior token for grounding across all test samples. Fig. 2(a) shows the hit ratio of four special tokens in each MLLM. Notably, the fixed token does not consistently exhibit the best grounding ability for different samples. For example, the highest hit ratio achieved by token '.' in LLaVA-1.5 is not more than $50\%$. In addition, for different MLLMs, the token at a fixed position does not always yield the best localization performance. For example, the

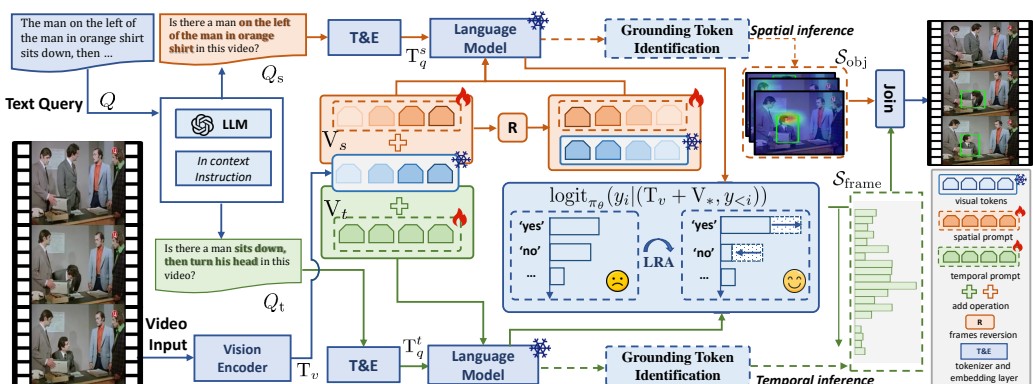

Figure 3: Overview of the proposed approach for zero-shot STVG. Given a video-text pair, we first decompose the text $Q$ into spatially and temporally related sub-queries, $Q_s$ and $Q_t$. The text prompt tokens converted from $Q_s$ and $Q_t$ are then concatenated with visual tokens $T_v$ for spatial and temporal inferences, respectively. In addition, we introduce learnable variables as visual prompts and optimize them by the logit-guided re-attention (LRA) module. For spatial grounding, we also develop a temporal-augmented assembling (TAS) strategy by reversing the frames to enhance temporal consistency. After optimization, we obtain the object track score $\mathcal{S}_{\text{obj}}$ and frame score $\mathcal{S}_{\text{frame}}$ based on the grounding token identification. The final prediction is derived by joining $\mathcal{S}_{\text{obj}}$ and $\mathcal{S}_{\text{frame}}$.

last special token ':', which is adopted in a previous work [69], obtains a high hit ratio in `Qwen-VL`, but its hit ratio is quite low in `LLaVA-1.5`. This observation holds for both image and video inputs.

**The special token with higher visual activation tends to show superior grounding performance.** Since the superior grounding tokens vary across different samples and MLLMs, identifying them for grounding is a problem that needs to be resolved when ground truth is unavailable as a prior. With further analysis, we reveal that the superior token for grounding tends to show higher visual activation. For each sample, we rank the special tokens according to the maximum value of visual attention and then evaluate their grounding accuracy by selecting the proposal with the highest attention value. Following the paradigm in previous works [50; 16], we extract box proposals using a detector and evaluate the grounding accuracy by the Acc@0.5 metric. Fig. 2(b) shows the results. We can see that the grounding accuracy decreases as the rank of visual activation reduces (from left to right). This supports our hypothesis that the special token with higher visual activation tends to show superior grounding ability. We have also observed that models with better comprehension possess better grounding ability overall. For example, the special tokens in `LLaVA-OneVision` achieve better grounding performance than those of the other MLLMs.

**A straightforward solution for zero-shot STVG.** Inspired by the grounding ability of special tokens subsequent to the text prompt, we refer to them as **_grounding tokens_** in our work. By identifying the special tokens characterized by high visual activation, we derive a strong training-free framework for STVG. Specifically, given the video-text pair $(V, Q)$, we first extract the object track proposals $\mathcal{O}_{pro} = \{O_1, \ldots, O_P\}$ from the video $V$, as done in previous works [3; 25], where $P$ denotes the number of proposals, and $O_p = \{b'_t\}_{t=1}^{t=T_v}$ is the set of bounding boxes of the $p$-th proposal in $T_v$ frames. Based on the foregoing findings, we then select the token with the highest attention value for locating the target object, and denote the text-to-image attention matrix of the selected token as $A_g \in \mathbb{R}^{1 \times M}$. As a result, we obtain each object track score by computing the maximum attention value inside each object track as:

$$\mathcal{S}_{\text{obj}} = \{s_1^o, s_2^o, \cdots, s_P^o\}, \text{ with } s_p^o = \max(A_g \odot f_{\text{B2M}}(O_p)), \quad (2)$$

where $\odot$ denotes element-wise multiplication. We choose the track with the highest score as spatial prediction $O'_{\text{pred}}$. In a similar manner, we can compute the frame score $\mathcal{S}_{\text{frame}} = \{s_1^t, s_2^t, \cdots, s_{T_v}^t\}$. By selecting the top-K frames with the highest scores from the temporal prediction $\mathcal{S}_{\text{frame}}$, we obtain the final spatio-temporal prediction $O'_{\text{pred}} = \{b'_t\}_{t=t_{s'}}^{t_{e'}}$, where $t_{s'}$ and $t_{e'}$ denote the starting and ending boundary predictions. Though simple, this solution has achieved comparable or even superior performance than current zero-shot methods. For example, based on `LLaVA-OneVision` model, this solution achieves 23.3% on the m_vIoU metric on HC-STVGv1 dataset, which outperforms previous SOTA result (19.1%) by E3M [3].

## 3.3 Decomposed Spatio-Temporal Highlighting

Despite a strong performance by the above solution, these grounding tokens often neglect some important cues during spatio-temporal localization especially when processing complex video queries. As shown in Fig. 1(b), the attribute cue *'in red clothes'* is overlooked during inference, which causes the incorrect spatial localization. We observe that the language query often contains attribute and action descriptions of the target object, which are beneficial for spatial and temporal localization, respectively. To this end, we propose a novel decomposed spatio-temporal highlighting (DSTH) strategy in our framework, which aims at highlighting the attributes/ actions cues in language query and enhances the spatial/temporal reasoning, respectively.

**Generation of target-related cues.** For comprehensive spatial and temporal reasoning, it is essential to extract the attribute and action descriptions from the original query $Q$ as textual cues of the target. Here, we leverage the strong in-context capability of the LLM [1] to extract attribute and action descriptions. Following previous works [62; 16], we construct a prompt with general instructions and in-context task examples. As shown in Fig. 3, we feed the prompt into LLM, and then these related descriptions $Q_s$ and $Q_t$ are generated through the LLM completion. More implementation details can be found in Appendix 7.3.

**Spatio-temporal prompt learning.** With the decomposed attribute and action descriptions as cues for spatial and temporal reasoning, the next question is how to efficiently direct the model to focus on the corresponding visual regions. Reliable responses in visual question answering (VQA) necessitate careful attention to the relevant visual context as pointed out by previous studies [58; 42]. Inspired by this, we innovatively propose regularizing the response to the questions constructed from the attribute and action descriptions for adjusting the visual attention of MLLMs. Specifically, we first transform the descriptions into interrogative queries by a fixed template to inquire the existence of the target. As shown in Fig. 3, from the original text input $Q$, we can obtain the attribute description *'a man on the left of the man in the orange shirt'*, which is further transformed into interrogative sub-query $Q_s$: *'Is there a man on the left of the man in the orange shirt in this video?'* for spatial inquiry. In a similar way, we can obtain the interrogative sub-query $Q_t$ for temporal inquiry. These interrogative sub-queries will be taken as input instructions of MLLMs for response generation.

**Logit-guided re-attention.** With extracted sub-queries above, we then propose a novel logit-guided re-attention (LRA) module to regularize the token prediction during response generation. Taking the spatial sub-query as an example, we first initialize a learnable variable $V_s$ with the same shape as the visual tokens $T_v$, and then add it to $T_v$ as the visual input of the language model. Sub-query $Q_s$ is converted as text prompt tokens $T_q^s$ by the text tokenizer and embedding layer. Given the input token sequence, the next token probability prediction over the vocabulary set $\mathcal{V}$ is formulated as:

$$p_y = \exp\left(\text{logit}_{\pi_\theta}(y_i|(T_v + V_s, T_q^s, y_{<i}))\right), \tag{3}$$

where $\pi_\theta$ denotes the parameter of the language model and is frozen in our work. $\text{logit}_{\pi_\theta}$ is the log probability of the generated token at time step $i$. $y_{<t}$ denotes the text tokens sequence prior to prediction time step $i$. We define the optimization objective by contrasting the probabilities of positive token *'yes'* and negative token *'no'*:

$$\mathcal{L}_s = 1 - \exp\left(\text{logit}_{\pi_\theta}(y_i^{yes}|(T_v + V_s, T_q^s, y_{<i})) - \text{logit}_{\pi_\theta}(y_i^{no}|(T_v + V_s, T_q^s, y_{<i}))\right). \tag{4}$$

During the inference process, we conduct backpropagation to optimize the learnable variable $V_s$ as the spatial prompt. The process is iterated $N_{ep}$ times by test-time tuning paradigm. By enhancing the positive response towards the sub-query $Q_s$, we can prompt the MLLM to effectively mine target-related contextual information during VQA, which in turn highlights the attribute cues in the original text. Similarly, we can obtain the temporal prompt $V_t$ by optimizing the temporal inference.

**Joint inference.** Based on the spatial and temporal visual prompts, we derive the attention maps $A_g^S$ and $A_g^T \in \mathbb{R}^{T_v \times h \times w}$ of the special token with high visual activation for spatial and temporal predictions, where $h$ and $w$ denote the token numbers of the height and width. From Eq. 2, we obtain the object track score $\mathcal{S}_{obj}$ and the temporal score $\mathcal{S}_{frame}$ based on $A_g^S$ and $A^T$, respectively. Finally, we integrate the predictions $O'_{pred} = \{b'_t\}_{t=t_{s'}}^{t_{e'}}$ as the spatio-temporal grounding result.

## 3.4 Temporal-augmented Assembling

The attribute sub-query, which provides a static state description, exhibits temporal independence for spatial grounding. In other words, the spatial grounding by the attribute sub-query should be

Table 1: Quantitative comparison on HCSTVG (v1&v2) and VidSTG (Declarative) benchmarks.

| Sup | Method | HCSTVG-v1 | | | HCSTVG-v2 | | | VidSTG (Declarative) | | |
|---|---|---|---|---|---|---|---|---|---|---|
| | | m_vIoU | vIoU@0.3 | vIoU@0.5 | m_vIoU | vIoU@0.3 | vIoU@0.5 | m_vIoU | vIoU@0.3 | vIoU@0.5 |
| Full | TubeDETR [59] [CVPR2022] | 32.4 | 49.8 | 23.5 | 36.4 | 58.8 | 30.6 | 30.4 | 42.5 | 28.2 |
| | STCAT [20] [NeurIPS2022] | 35.0 | 57.7 | 30.0 | – | – | – | 33.1 | 46.2 | 32.6 |
| | CSDVL [33] [CVPR2023] | 36.9 | 62.2 | 34.8 | 38.7 | 65.5 | 33.8 | 33.7 | 47.2 | 32.8 |
| | CG-STVG [15] [CVPR2024] | 38.4 | 61.5 | 36.3 | 39.5 | 64.5 | 36.3 | 34.0 | 47.7 | 33.1 |
| Weak | WINNER [29] [CVPR2023] | 14.2 | 17.2 | 6.1 | – | – | – | 11.6 | 14.1 | 7.4 |
| | VEM [21] [ECCV2024] | 14.6 | 18.6 | 5.8 | – | – | – | 14.5 | 18.6 | 8.8 |
| | CoSPaL [25] [ICLR2025] | 22.1 | 31.8 | 19.6 | 22.2 | 31.4 | 18.9 | 16.0 | 20.1 | 13.1 |
| | STPro [13] [CVPR2025] | 17.6 | 27.0 | 12.9 | 20.0 | 31.1 | 14.6 | 15.5 | 19.4 | 12.7 |
| ZS | RedCircle [47] [CVPR2023] | 9.2 | 7.8 | 1.6 | – | – | – | 8.6 | 7.6 | 0.9 |
| | ReCLIP [50] [ACL2022] | 14.4 | 18.3 | 4.9 | – | – | – | 14.2 | 17.5 | 7.9 |
| | E3M [3] [ECCV2024] | 19.1 | 29.4 | 10.6 | – | – | – | 16.2 | 20.5 | 11.9 |
| | Ours LLaVA-Next-Video-7B | 20.4 | 33.6 | 12.4 | 23.6 | 36.8 | 15.5 | 16.6 | 26.8 | 11.1 |
| | Ours ShareGPT4Video-8B | 20.0 | 32.2 | 10.9 | 24.4 | 38.9 | 15.4 | 17.1 | 27.8 | 11.6 |
| | Ours Qwen2-VL-7B | 23.6 | 39.0 | 14.4 | 25.6 | 40.5 | 17.1 | 17.0 | 27.4 | 11.4 |
| | Ours LLaVA-OneVision-7B | 24.8 | 41.5 | 16.3 | 27.7 | 44.7 | 19.5 | 18.0 | 29.8 | 12.2 |

temporally consistent for temporal augmentation (*e.g.*, reversing the order of video frames). However, there exists temporal inconsistency when introducing temporal augmentation in current MLLMs. Here, we propose a metric to measure the temporal consistency. By denoting the spatial attention maps before and after reversing the order of input frames as $A_{\mathrm{g}}^{S}$ and $\tilde{A}_{\mathrm{g}}^{S}$, the temporal consistency can be measured as:

$$S_{\mathrm{cons}} = \max\{s_1, \cdots, s_P\}, \quad s_p = \max\left((A_{\mathrm{g}}^{S} \odot f_{\mathrm{B2M}}(O_p)) \odot (\tilde{A}_{\mathrm{g}}^{S} \odot f_{\mathrm{B2M}}(O_p))\right), \quad (5)$$

where a higher $S_{\mathrm{cons}}$ indicates better temporal consistency. We then divide the testing samples into ten groups in descending order of temporal consistency. Fig. 4 shows the average grounding accuracy of each group samples. The results demonstrate a pronounced association between temporal consistency and spatial grounding performance. The temporal inconsistency tends to cause worse spatial localization.

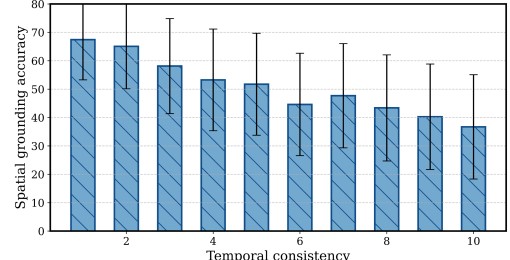

Figure 4: Spatial grounding accuracy of different groups of samples on the HC-STVGv1 dataset. These groups are ranked by descending temporal consistency.

To this end, we integrate a temporal-augmented assembling (TAS) strategy in our framework. As shown in Fig. 3, we perform a frame-level reversion operation on the visual tokens and the spatial prompt simultaneously, and then optimize the spatial prompts for the input of the original frames and the temporal-augmented input, respectively. During inference, we derive the spatial prediction by assembling the attention maps of temporal-augmented input frames. The proposed TAS strategy alleviates the effect of temporal inconsistency and improves the robustness of spatial grounding.

## 4 Experiments

### 4.1 Settings

**Datasets.** We evaluate on three video benchmark datasets: HCSTVG-v1, HCSTVG-v2 [51], and VidSTG [73]. We provide detailed introduction about them in Appendx 7.1.

**Implementation Details.** We adopt G-DINO [38] and SAM2 [45] for detection and tracking tubelet generation, and use GPT-4o to decompose the original query sentence into spatial and temporal sub-queries. We consider four widely-used video MLLMs: LlaVA-Next-Video-7B [27], Qwen2-VL-7B [53], ShareGPT4Video-8B [7], LlaVa-OneVision-7B [27] for demonstrating the efficiency of our method. Refer to Appendix 7.2 for more details.

**Evaluation Metrics.** We follow the standard evaluation protocol [59; 25] and use $m\_vIoU$, and $vIoU@R$ to assess the performance of spatio-temporal grounding. Specifically, let $S_i$, $S_u$ denote the intersection and union between the predicted and ground-truth frames. The $vIoU$ is computed by $\frac{1}{S_u} \sum_{t \in S_i} \mathbf{IoU}(b'_t, b_t)$, where $b'_t$ and $b_t$ denote the detected and ground-truth bounding box at frame $t$, respectively. The $m\_vIoU$ represents the $vIoU$ averaged over all testing samples, and $vIoU@R$ denotes the proportion of data samples in the testing subset with $vIoU$ greater than the threshold $R \in \{0.3, 0.5\}$.

## 4.2 Performance Comparison

**Quantitative Comparison.** Tab. 1 presents a comparison of our method against 11 methods from three categories, including zero-shot, weakly-supervised, and fully-supervised methods. Specifically, we compare our approach with existing zero-shot SOTA approaches (*e.g.*, E3M [3], ReCLIP [50], RedCircle [47]). Our method consistently outperforms these methods by a remarkable margin on all benchmarks. Based on `LLaVA-Next-Video-7B`, our method outperforms E3M by $4.2\%$ on $vIoU@0.3$ and $1.8\%$ on $vIoU@0.5$. When integrated into the better `LLaVA-OneVision-7B` model, the corresponding improvements reach $12.1\%$ and $5.7\%$.

This shows that our framework can adapt to various MLLMs well, and better performances can be achieved by using better MLLMs. Even on the VidSTG dataset, which contains fewer action cues related to the target and thus makes temporal grounding particularly challenging, our framework still outperforms the previous SOTA methods overall. This demonstrates the strong generalization capability of our method. Besides, our method even surpasses current SOTA weakly-supervised methods on most metrics. For example, on the HCTSVG-v2 benchmark, our method outperforms CoSPaL [25] by a margin of

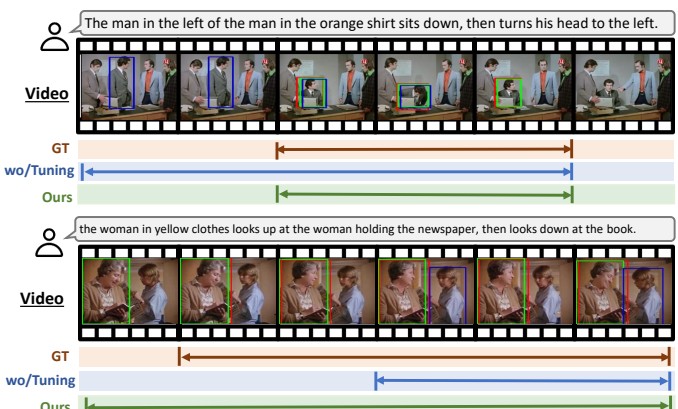

Figure 5: Qualitative results on the HC-STVGv1 test set. Better spatio-temporal grounding results (green) are obtained when the DSTH strategy is being used for optimization.

$5.5\%$ on the $m\_vIoU$ metric. Furthermore, compared to the fully-supervised methods, our method can still achieve comparable results, which further validates the superiority of our approach. We provide quantitative comparison on the VidSTG (Interrogative) and image benchmarks in Appendix 8.1

**Qualitative Comparison.** We present qualitative results in Fig. 5. In the example below, before introducing the DSTH optimization strategy, the model neglects the attribute cues in the language query and suffers from the spatial grounding error. By highlighting the attribute cues of the target, our method can direct the MLLMs toward reliable visual context and improve spatial localization.

## 4.3 Ablation Study

In this section, based on HC-STVGv1 dataset, we analyse the effect of different proposed components when integrated into `LlaVa-Next-Video` and `LlaVa-OneVision`. We also conduct extensive ablation experiments with the hyper-parameters based on `LlaVa-Next-Video`.

**Component analysis.** In Tab. 2, we first average the attention maps of all special tokens (*1st* row) as the baseline. This solution has achieved outstanding performance. Next, in the *2nd* row, we integrate the selection of the superior token introduced in grounding token identification (GTI). It brings consistent improvements on different MLLMs. The results show that identifying the superior special token can effectively unleash the powerful comprehension ability of MLLMs. We them validate the efficiency of the decomposed spatio-temporal highlighting strategy (*3rd* and *4th* rows). $\mathbf{S}_{\text{tune}}$ and $\mathbf{T}_{\text{tune}}$ denote adopting the prompt learning in spatial and temporal inferences, respectively. It is demonstrated that the MLLMs can be directed to focus on the spatial/temporal related regions better

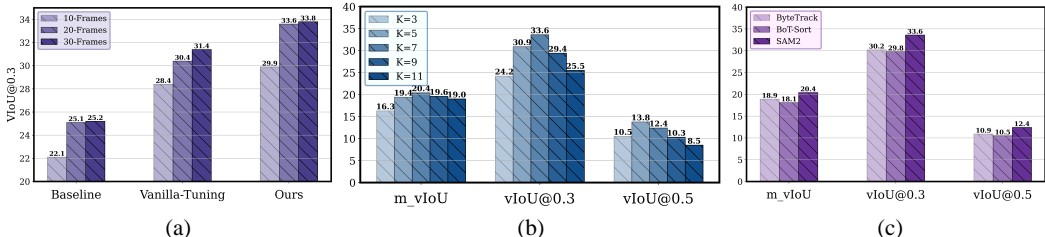

Figure 6: (a) Comparison on the number of frames $N_f$ as input, using vIoU@0.3. (b) Ablation on the number of selected frames K during temporal prediction. (c) Ablation on different trackers.

when highlighting the attribute/action cues with proper prompts. When utilizing prompt learning on the two sub-tasks, the final performance can be further refined (**5th** row). In addition, we also find that the DSTH strategy can improve the grounding performance especially for the less efficient MLLMs (*e.g.*, `LLaVA-Next-Video`). Even for the MLLMs (*e.g.*, `LLaVA-OneVision`) with strong temporal comprehension ability, performance improvement can still be achieved by the test-time optimization.

Finally, we introduce the temporal-augmented assembling (TAS) strategy (**6th** row). The overall performance can be further improved by refining the spatial localization.

Table 2: Component ablation on `LLaVA-Next-Video` and `LLaVA-OneVision`.

| GTI | $S_p$ | $T_p$ | TAS | LLaVA-Next-Video | | | LLaVA-OneVision | | |
|---|---|---|---|---|---|---|---|---|---|
| | | | | m_vIoU | vIoU@0.3 | vIoU@0.5 | m_vIoU | vIoU@0.3 | vIoU@0.5 |
| ✗ | ✗ | ✗ | ✗ | 15.2 | 25.1 | 8.5 | 21.3 | 36.1 | 12.6 |
| ✓ | ✗ | ✗ | ✗ | 16.3 | 26.6 | 9.3 | 23.3 | 38.8 | 15.1 |
| ✓ | ✓ | ✗ | ✗ | 18.0 | 30.1 | 10.1 | 24.1 | 40.3 | 16.0 |
| ✓ | ✗ | ✓ | ✗ | 18.4 | 29.5 | 10.1 | 23.8 | 39.7 | 15.5 |
| ✓ | ✓ | ✓ | ✗ | 19.9 | 32.1 | 11.9 | 24.3 | 40.7 | 16.0 |
| ✓ | ✓ | ✓ | ✓ | 20.4 | 33.6 | 12.4 | 24.8 | 41.5 | 16.3 |

**Why does spatial and temporal prompt learning by contrasting logits of *'yes'* and *'no'* work.** Though contrasting logits of *'yes'* and *'no'* tokens is heuristic, it is reasonable. We empirically find that given the question prompt, the model often gives ambiguous or even wrong response predictions even though the objects referred to by the text do indeed exist in the video. This indicates that the model cannot understand the video content well in certain cases. By optimizing the logit of the *'yes'* token toward positive prediction instead of the *'no'* token, we encourage the model to carefully attend to the target-related visual information and achieve better visual localization.

To further validate the hypothesis, we measure the distribution of attention peaks before and after test-time optimization. Based on the evaluation on the HC-STVG dataset, we find that the average attention peak before test-time optimization is about 6.7 (without normalization), and after optimization, it increases to 23.9 (without normalization). This change in attention values indicates that our proposed optimization strategy can encourage the model to perceive more visual modal information for generating visually-grounded responses, thereby justifying our previous hypothesis.

**Comparison of different learning strategies.** We propose logit-guided re-attention by contrasting logits of *'yes'* and *'no'* tokens. Here, we also consider entropy minimization (i.e., increasing the predicted probability of the *'yes'* token across the entire vocabulary) during test-time tuning. The results obtained by `LLaVa-Next-Video-7B` on the HC-STVG-v2 dataset are shown in the table below.

The **1st** row denotes the results of introducing grounding token identification (GTI). The **2nd** and **3rd** rows denote the test-time tuning results achieved by entropy minimization and contrasting the logits of *'yes'* and *'no'*, respectively. We find that entropy minimization does not bring noticeable improvement. We believe this is be-

Table 3: Comparison of different learning strategies.

| Methods | m_vIoU | vIoU@0.3 | vIoU@0.5 |
|---|---|---|---|
| GTI | 19.2 | 28.5 | 12.4 |
| GTI + entropy minimization | 19.5 | 28.5 | 12.5 |
| GTI + contrasting the logits | 22.1 | 32.9 | 14.0 |

cause entropy minimization may be achieved by reducing the token prediction of abundant irrelevant tokens in the vocabulary while ignoring the differentiation between the *'yes'* and the *'no'* token. As a result, it is less efficient than explicitly contrasting the logits in promoting the model to attend to abundant visual information.

**Generality analysis of our method.** We conduct the scalability verification on `InternVL3` and `Qwen2.5-VL` series models. With limited computational resources, we evaluate our method on `InternVL3-4B`, `InternVL3-8B`, `Qwen2.5-VL-3B` and `Qwen2.5-VL-7B` models in the table below.

We average the attention maps of all special tokens as the baseline. Benefiting from our grounding token identification and test-time optimization, we can obviously achieve better grounding performance not only in the smaller but also the larger models, which well demonstrates both the generality and scalability of our method. Moreover, we find that the model size significantly impacts the inherent grounding ability of special tokens. The performance of small models (*e.g.*, `InternVL3-4B`) is inferior to the larger models (*e.g.*, `InternVL3-8B`). It is because fewer parameters in language models perform poorly in multimodal alignment.

Table 4: Evaluation on different scales of models.

| Methods | vIoU@0.3 | m_vIoU | t_iou | gt_viou |
|---|---|---|---|---|
| `InternVL3-4B` + Baseline | 24.5 | 17.0 | 38.4 | 38.2 |
| `InternVL3-4B` + Ours | 41.1 | 25.4 | 45.5 | 49.5 |
| `InternVL3-8B` + Baseline | 36.5 | 22.9 | 42.9 | 46.1 |
| `InternVL3-8B` + Ours | 45.3 | 27.5 | 47.2 | 53.2 |
| `Qwen2.5-VL-3B` + Baseline | 28.5 | 17.3 | 37.1 | 39.1 |
| `Qwen2.5-VL-3B` + Ours | 41.7 | 25.4 | 45.9 | 49.3 |
| `Qwen2.5-VL-7B` + Baseline | 39.7 | 23.6 | 44.1 | 47.2 |
| `Qwen2.5-VL-7B` + Ours | 45.4 | 27.7 | 46.9 | 51.6 |

**Effect of input frames** $N_f$**.** Fig. 6(a) analyzes the effect of using different numbers of video frames $N_f$ as input. We can see that more input frames provide richer visual context for model inference and lead to a better overall performance. Besides, the performance becomes slightly peaked as the number of input frames increases. To balance between performance and efficiency during inference, we uniformly sample 20 frames as the visual input of MLLMs. In Fig. 6, we also evaluate `Vanilla-Tuning`, a solution that directly applies test-time optimization to the entire text query $Q$ instead of decomposing it into attribute/action sub-queries. We observe that the performance of `Vanilla-Tuning` is inferior to our method. It demonstrates that decomposing the text query into attribute/action sub-queries can help facilitate the intensive spatio-temporal comprehension of MLLMs.

**Ablation on the number** $K$ **of predicted frames.** Fig. 6(b) analyzes the effect of predicted frame numbers $K$ for temporal grounding based on the HCSTVG-v1 dataset. Here, we consider selecting the optimal frame number from the set $\{3, 5, 7, 9, 11\}$. In general, when $K$ is set as 7, the optimal result can be obtained. Thus, we select top-7 frames as the prediction during temporal grounding.

**Trackers ablation.** Fig. 6(c) analyzes the effect of different trackers. Besides SAM2 [45], a foundation model for tracking, we also consider two other tracking models (*i.e.*, ByteTrack [72] and BoTSort [2]) for analysis. It is reasonable that when a stronger tracking model (*e.g.*, SAM2) is used for generating spatio-temporal tubelets, we can obtain better STVG performances. Besides, when suboptimal tracking models are used, our method can still achieve comparable or better performances than the current SOTA methods, which shows the generalization of our approach.

## 5   Conclusion

In this paper, we have presented a novel MLLM-based zero-shot framework for the spatio-temporal video grounding (STVG) task. Our approach is initiated by identifying the grounding capability of special tokens in widely used MLLMs during response generation. To leverage and unleash the comprehension ability of MLLMs, we have proposed the decomposed spatio-temporal highlighting (DSTH) strategy. It first decomposes the text query into attributes and actions sub-queries. It then employs a logit-guided re-attention (LRA) module to sharpen the spatial/temporal visual context comprehension. We have also proposed the temporal-augmented assembling (TAS) strategy to alleviate the effect of temporal inconsistency. Extensive experiments conducted on three STVG benchmarks demonstrate the effectiveness of our proposed framework.

Our approach does have limitations. For example, it may struggle to process long videos well due to high computational consumption caused by MLLMs. As a future work, we would like to consider incorporating token pruning and key frame selection techniques into the model design.

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

# Appendices of Unleashing the Potential of Multimodal LLMs for Zero-Shot Spatio-Temporal Video Grounding

# 6 Grounding Tokens

## 6.1 Illustration of Input Tokens Components

In Fig. 7, we give a clear illustration of input token components in MLLMs. In practice, besides the visual and text prompt tokens (② and ③), there are also system tokens and some special tokens (① and ④) equipped for language generation. In particular, the grounding ability of special tokens subsequent to the text instruction prompt is significantly undervalued in previous works. Here we focus on exploring the grounding ability of these special tokens. For different multi-modal large language models, there often are different special tokens due to different instruction-tuning process. For example, in `Llava-1.5`, the introduced special tokens include: {':', 'ANT', 'IST', 'SS', '_A'}. While for the `Qwen2-VL`, these special tokens include: { Ċ, 'assistant', '<|im_start|>', '<|im_end|>'}. Note that we also consider the '.' as one of the special tokens considering that it is located in the end of the instruction prompt and is characteristic of sink tokens [23; 57].

In fact, special tokens are predefined symbols within a language model's vocabulary. They focus on guiding the model's processing instead of representing real words and guide the model to generate coherent and context-aware responses. In the scenario explored in our work (*i.e.*, dialogue systems), special tokens can differentiate between a user's question and the assistant's answer. By using special tokens, models become better at understanding structure and context.

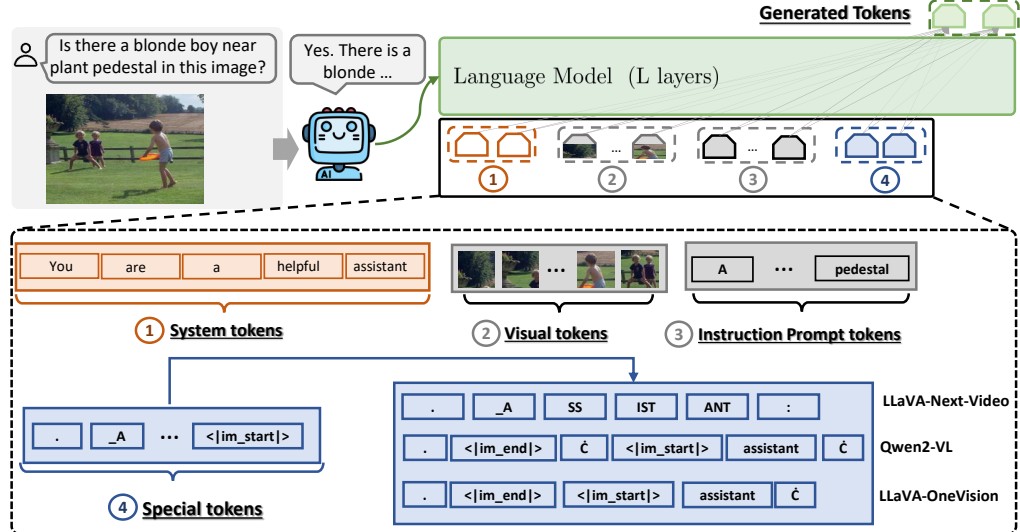

Figure 7: Illustration of the tokens input in MLLMs-based dialog system. Here, we take the image question answering as an example.

## 6.2 Quantitative Analysis of Special Tokens

In our work, we randomly selected a subset of 1000 image-text pairs from RefCOCOg [17] validation set for image MLLMs analysis, and a subset of 1000 video-text pairs from HC-STVG [51] dataset for video MLLMs analysis. Particularly, we choose six typical MLLMs (*i.e.*, LLaVA-1.5, Qwen-VL, Deepseek-VL, LLaVA-Next-Video, Qwen2-VL, LLaVA-OneVision) for pilot studies. In the following, we will introduce our findings about special tokens in MLLMs.

**Some special tokens show outstanding grounding ability for text prompt input.** We compare the grounding accuracy of G-DNIO [38] and the special tokens' ones by probing the attention map. Here we adopt the G-DNIO with Swin-T backbone and it is not finetuned on the refcoco series datasets. Following previous visual grounding works [50; 16], we adopt the IoU@0.5 as the metric of grounding accuracy. The results evaluated on refcoco series benchmarks are shown in the Fig. 8. The yellow horizontal line denotes the result by G-DINO model. Though the G-DINO is trained on image-text pairs in fully-supervised manner, some special tokens still achieve comparable and even

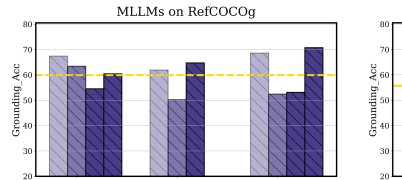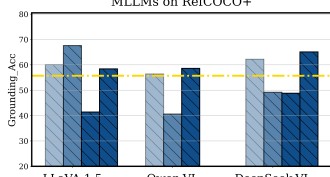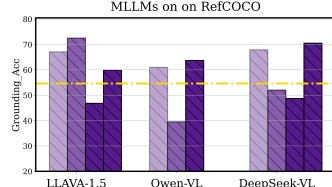

Figure 8: Comparison of grounding accuracy between G-DINO and special tokens on RefCOCO series datasets.

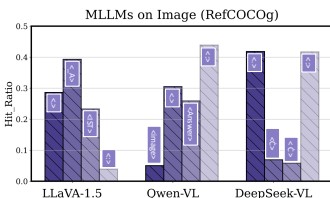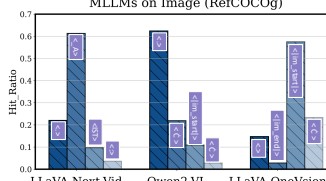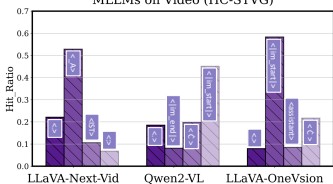

Figure 9: The hit ratio of different special tokens in image and video MLLMs.

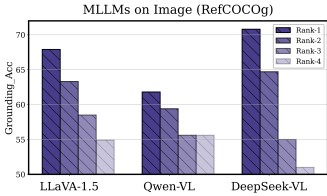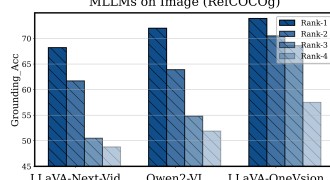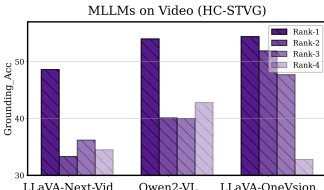

Figure 10: Grounding accuracy of special tokens ranked by the visual activation degree.

better performance. The results show that the special token can effectively ground the text prompt input by integrating the textual cues. Besides, we also notice that there exists a pronounced difference for grounding ability of different special tokens.

**MLLMs dynamically assign the special tokens to attend to the text-related regions.** For further exploration on the grounding ability of special tokens, we define the ***attention ratio*** of each special token as the ratio of maximum attention within the ground-truth bounding box $b_{gt}$ to that outside it. Here, a higher ratio indicates better target grounding ability. Given a test sample, we can identify the token that yields the highest attention ratio as the superior token for grounding. Then a token's ***hit ratio*** is defined as the frequency of being the superior token for grounding across all test samples. The Fig. 9(a) shows the hit ratio of special tokens in each MLLM. We choose four special tokens for visualization. In addition to the findings that MLLMs dynamically assign the special tokens to attend to the text-related regions, we also observe that the superior token for grounding in video MLLMs shows a more concentrated trend than image MLLMs. For example, the highest hit ratio by `Qwen2-VL` is more than $60\%$. Besides, for the same special token in the MLLM, the hit ratio in the case of different datasets may be significantly different. For example, the token '.' shows a high hit ratio in RefCOCOg dataset, but its hit ratio is quite low in the HC-STVG dataset.

**The special token with higher visual activation tends to show superior grounding performance.** In our work, with further analysis, we reveal that the superior token for grounding tends to show higher visual activation. For each sample, we rank the special tokens according to the maximum value of visual attention and evaluate their grounding accuracy by selecting the proposal with the highest attention value as the prediction. In practice, we extract box proposals by the detector (*e.g.*, G-DINO) and evaluate the grounding accuracy by the Acc@0.5 metric. Fig. 10 shows the results. We can see that the grounding accuracy decreases as the rank of visual activation reduces (from left to right). We also observe that models with better comprehension overall possess better grounding ability.

## 6.3 Qualitative Analysis of Special Tokens

In Fig. 11, we visualize token-to-visual attention maps using some image-text pairs by LLaVA-1.5-7B model. In particular, for each image-text pair, we visualize the visual attendance of the semantic tokens from the text prompt and special tokens. The green bounding box denotes the ground truth of the target object. Notably, we notice that the semantic tokens in text prompts can provide tangible attention to related visual entities by the limited cues, while the special tokens are often assigned to integrate global instruction cues and attend to the exact target region. For example, given the text prompt *'a man getting ready to cut a cake'*, the token 'man' and 'cake' both show reasonable visual response. The special token '_A' can accurately attend to the target person. In addition, we see that the special token with hight visual activation often shows better grounding performance. However, we also see that the special tokens may attend to the wrong instance even though the semantic tokens can properly capture their region of interest. It suggests that there is still room for improvement in inference in current multimodal language models.

In Fig. 12, we visualize attention maps of special tokens using some video-text pairs by ShareGPT4Video-8B model. The ground-truth spatio-temporal tube is denoted by red bounding boxes. Each row shows the attention maps of a specific token. We find that only a few special tokens can well capture the corresponding target regions, which necessitates the efficient selection of grounding tokens. Also, in some cases (*e.g.*, the below sample), there may be more than one special token that will pay attention to the target area. This shows that achieving localization by considering the integration of multiple effective special tokens can be a direction for improvement. We leave it to future research.

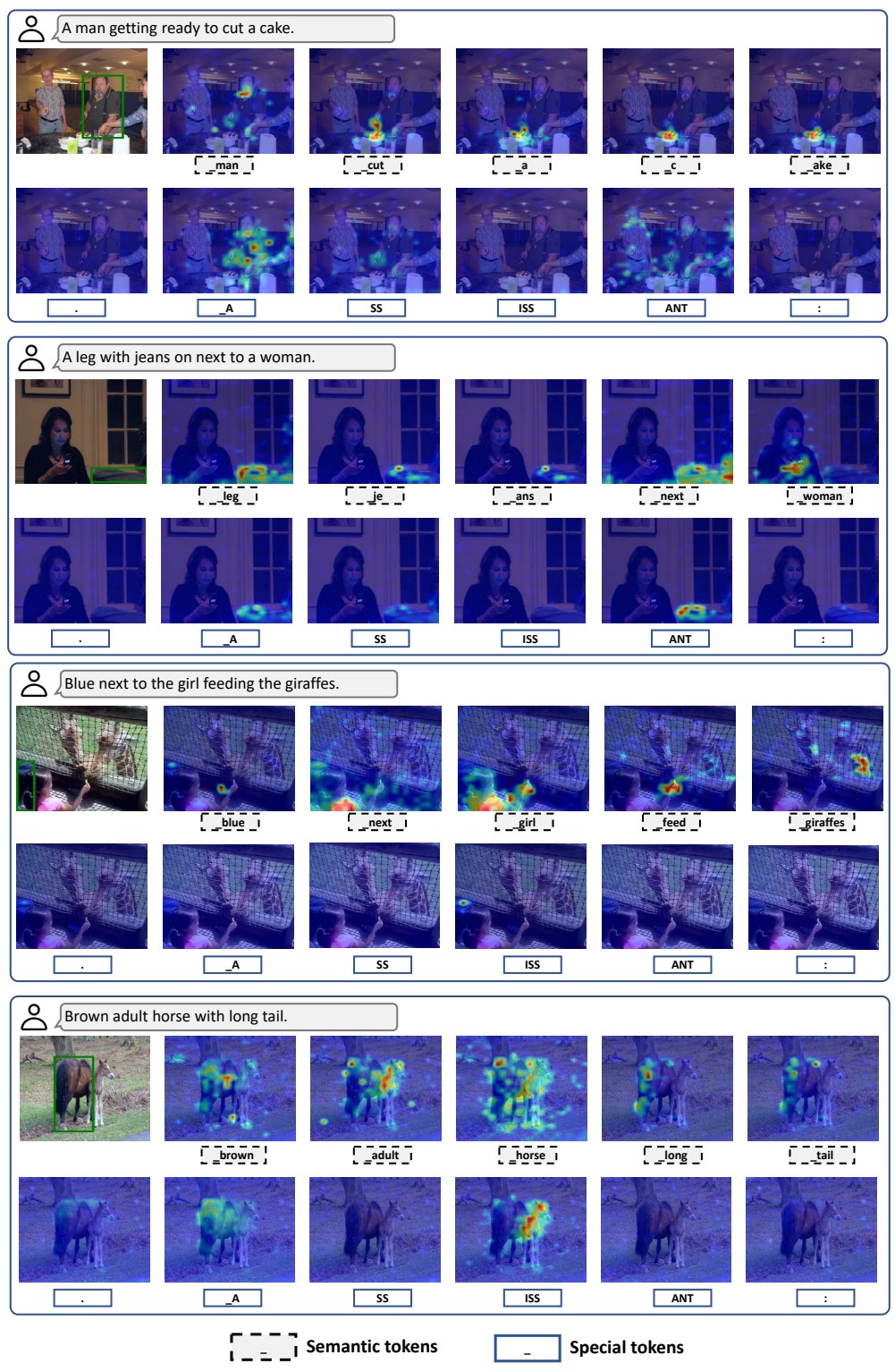

Figure 11: Visualization of tokens' attention maps on image-text pairs. The tokens with dotted box denote the semantic tokens in text prompts while the tokens with solid box denote the special tokens following the text prompts.

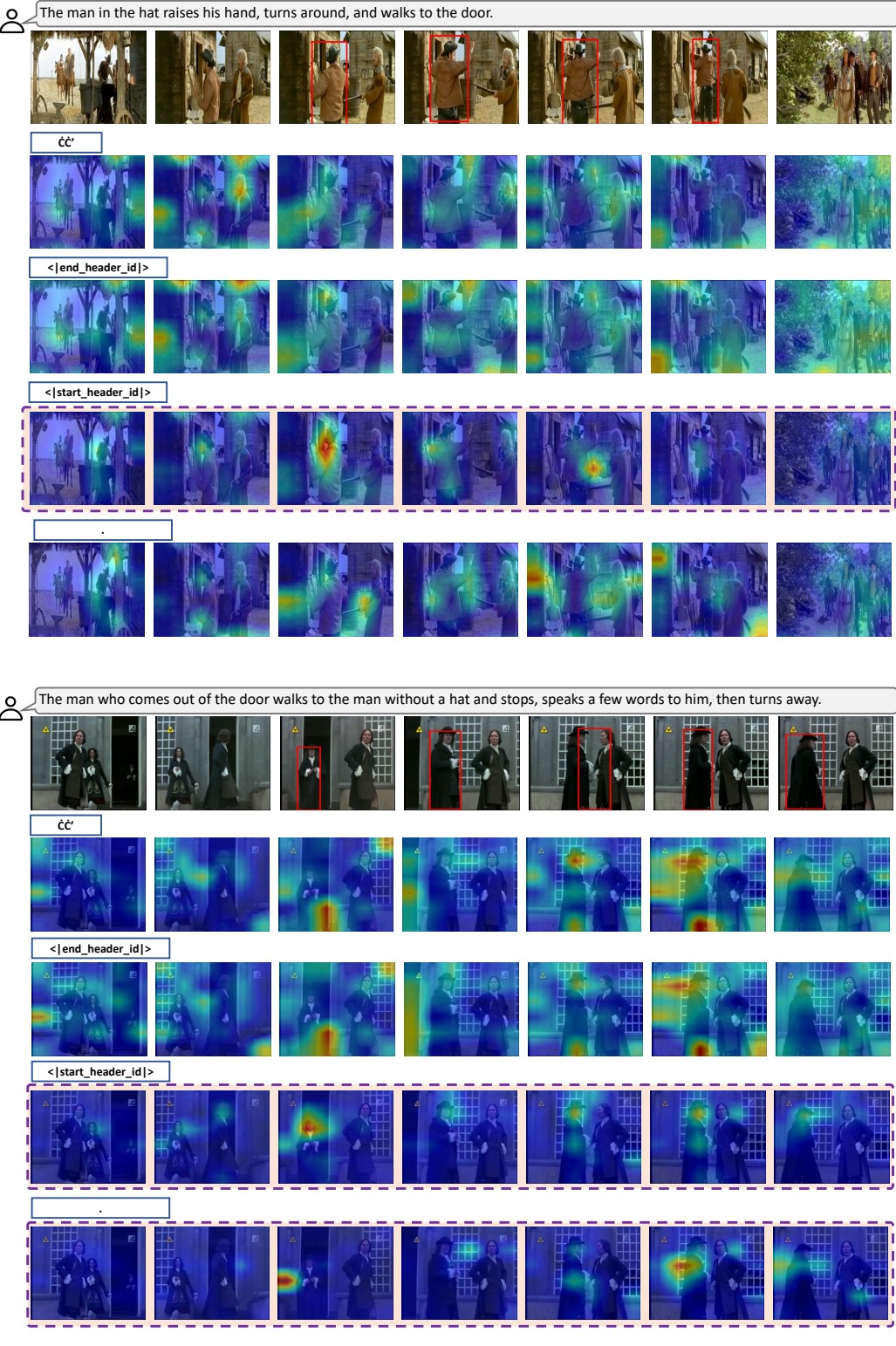

Figure 12: Visualization of tokens' attention maps on video-text pairs.

# 7 More Implementation Details

## 7.1 Datasets

We evaluate on three video benchmark datasets: HCSTVG-v1, HCSTVG-v2 [51], and VidSTG [73]. HCSTVG-v1 has 4500 training and 1160 testing videos with sentence descriptions of human attributes/actions. HCSTVG-v2 extends v1 to 16,544 videos, including 10,131 training, 2,000 validation, and 4,413 testing videos. Since the test set is unavailable, we evaluate on the validation set following prior works [59; 25]. VidSTG includes 99,943 video-sentence pairs (44,808 declarative, 55,135 interrogative), covering 10,303 videos and 80 object categories. Training, validation, and test sets have 80,684, 8,956, and 10,303 pairs, respectively.

## 7.2 Method Implementation

In this work, we adopt G-DINO [38] with $0.4$ for both phrase and box thresholds to detect object proposals, and then utilize SAM2 [45] as tracker for tubelet proposals generation. Considering the information redundancy, we run the detector every 10 frames for efficiency. We utilize GPT-4o to decompose the original query sentence into spatial and temporal sub-queries. To demonstrate the efficiency of our method, we consider four LLaVA-like video MLLMs: `LLaVA-Next-Video-7B` [27], `Qwen2-VL-7B` [53], `ShareGPT4Video-8B` [7], `LLaVa-OneVision-7B` [27]. In practice, with the limited tokens context length and computing efficiency, we sample 20 frames by default as the visual input of MLLMs for each video. When adopting the test-time tuning strategy DSTH, we set the learning rate lr and iteration times $N_{ef}$ as $8.0$ and 1 according to the ablation analysis in Tab. 7. We conduct all experiments on an A100 GPU with 80G VRAM based on Pytorch framework. To better capture text-to-visual attention, we introduce *'Describe this video in details.'* as general query prompt $Q_{\text{cap}}$ and implement the relative attention strategy [69] to reduce the effect of visual registers [9; 55]. In Algo. 1, we outline the implementation of the proposed decomposed spatio-temporal highlighting strategy during test-time tuning.

## 7.3 Generation of Target-related Cues

In this work, we extract the attributes and actions descriptions from the original query $Q$ as textual cues to enhance the spatial and temporal comprehension, respectively. To obtain multiple target-related cues, we leverage the strong in-context capability of the Large Language Model (LLM). In particular, we construct the in-context instruction to prompt llm for completion. The whole prompt used in this work includes:general instruction (in brown), output constraints (in blue), in-context task examples (in green) and input question (in yellow), shown in Fig. 13.

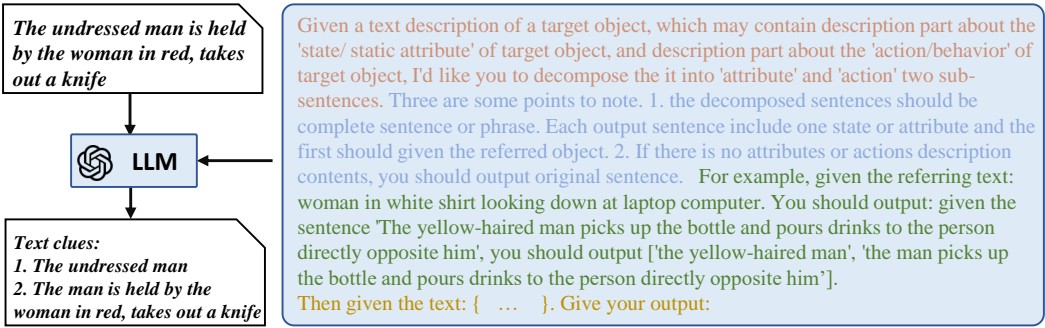

Figure 13: Flow of LLM-based generation of target-related cues.

In Fig. 14, we present some examples of LLM generation. After decomposing the original text description into attributes and actions-related descriptions, we will further transform the descriptions into interrogative queries by a fixed template to inquire about the existence of the target. In our work, we adopt the template *'Is there a ___ in this video?'*.

- Original Text: The man in purple clothes takes a step forward and sits on the bench.
- Attributes Text : the man in purple clothes.
- Actions Text : The man takes a step forward and sits on the bench.

10_phVLLTMzmKk.mp4

- Original Text: The yellow-haired man takes the opposite woman's hand and kisses it, then puts it down.
- Attributes Text : the yellow-haired man.
- Actions Text : The man takes the opposite woman's hand and kisses it, then puts it down.

159_vsMgg4snZzM.mkv

- Original Text: The white-bearded man points at the man on his right and then points at himself again.
- Attributes Text : the white-bearded man.
- Actions Text : The man points at the man on his right and then points at himself again.

111_pSdPmmJ3-ng.mp4

Figure 14: Examples of LLM-based cues generation.

---

**Algorithm 1:** Decomposed Spatio-Temporal Highlighting

---

**Input:** MLLM $\pi_\theta$, video query input $Q$, video $X$, track proposals $O_{\mathrm{pro}} = \{o_1, \ldots, o_P\}$, test-time tuning epochs $\mathrm{N}_{ep}$

**Output:** Generated tube $O'_{\mathrm{pred}} = \{b_t\}_{t=t_{s'}}^{t_{e'}}$ based on query input $Q$ and video $X$

Decompose the $Q$ into attribute/action sub-queries $Q_\mathrm{s}$ and $Q_\mathrm{t}$ for spatial/temporal inquiry.

Initialize spatial prompt $V_s$ and temporal prompt $V_t$.

Initialize $i_{ep} = 0$.

Embed the video $X$ into visual tokens $\mathrm{T}_v$. Embed the $Q_\mathrm{s}$ and $Q_\mathrm{t}$ into text tokens $\mathrm{T}_q^s$ and $\mathrm{T}_q^t$.

**while** *training* **do**

    **while** $i_{ep} < \mathrm{N}_{ep}$ **do**

        `// spatial prompt Learning`

        Compute positive and negative logit prediction for attribute subquery $Q_\mathrm{s}$:

        $\mathrm{p}^{yes} = \mathrm{logit}_{\pi_\theta}(y_i^{yes}|(\mathrm{T}_v + \mathrm{V}_s, \mathrm{T}_q^s, y_{<i})), \mathrm{p}^{no} = \mathrm{logit}_{\pi_\theta}(y_i^{no}|(\mathrm{T}_v + \mathrm{V}_s, \mathrm{T}_q^s, y_{<i}))$

        Update $\mathrm{V}_s$ by minimizing $\mathcal{L}_\mathrm{s} = -\exp(\mathrm{p}^{yes} - \mathrm{p}^{no})$.

        `// temporal prompt Learning`

        Compute positive and negative logit prediction for action query $Q_\mathrm{t}$:

        $\mathrm{p}^{yes} = \mathrm{logit}_{\pi_\theta}(y_i^{yes}|(\mathrm{T}_v + \mathrm{V}_t, \mathrm{T}_q^t, y_{<i})), \mathrm{p}^{no} = \mathrm{logit}_{\pi_\theta}(y_i^{no}|(\mathrm{T}_v + \mathrm{V}_t, \mathrm{T}_q^t, y_{<i}))$

        Update $\mathrm{V}_t$ by minimizing $\mathcal{L}_\mathrm{t} = -\exp(\mathrm{p}^{yes} - \mathrm{p}^{no})$.

        $i_{ep} = i_{ep} + 1$.

    **end**

**end**

prepare general query prompt $Q_{\mathrm{cap}}$.

**while** *inference* **do**

    cache the visual attention map $A_{\mathrm{cap}}$ by query $Q_{\mathrm{cap}}$ as input.

    `// inference with` $\mathrm{T}_v + \mathrm{V}_s$.

    cache the visual attention map $A_\mathrm{g}^S$ by query $Q$ with $\mathrm{T}_v + \mathrm{V}_s$ as visual tokens.

    compute spatial-related visual attention map $A_\mathrm{g}^S = A_\mathrm{g}^S / A_{\mathrm{cap}}$,

    obtain the object track prediction.

    `// inference with` $\mathrm{T}_v + \mathrm{V}_t$.

    cache the visual attention map $A_\mathrm{g}^T$ by query $Q$ with $\mathrm{T}_v + \mathrm{V}_t$ as visual tokens.

    compute temporal-related visual attention map $A_\mathrm{g}^T = A_\mathrm{g}^T / A_{\mathrm{cap}}$,

    obtain the target frames prediction.

    Combine the object track prediction and target frames prediction.

**end**

---

# 8 More Results and Ablation Analysis

## 8.1 Comparison on VidSTG (Interrogative) and RefCOCO Benchmarks

In Tab. 5, we compare our approach with other SOTA methods on the VidSTG (Interrogative) benchmarks. Even given the interrogative sentence as query, our method still show superior performance with a 4.0% improvement on the vIoU@0.5 metric when integrated with `LLaVA-Next-Video-7B` model, which show the strong generalization capability of our framework. Undoubtedly, our method can also achieve superior localization performance in the image grounding tasks. In Tab. 6, we compare our framework with other SOTA visual grounding methods on RefCOCO series benchmarks. Here we just introduce the selection of superior token for grounding and has achieved comparable and even better performance than current zero-shot SOTA methods. The The 'Oracle' denotes the result of choosing the most exact one of candidate boxes (extracted by G-DINO) with knowledge of the ground truth. We also show the proportion of our method's performance relative to oracle's, which is indicatd by the number in lower right corner. Note that our insight about grounding tokens is orthogonal to other works [69; 30; 24]. We believe that it can be integrated with other works to achieve better results.

Table 5: Comparison on VidSTG (Interrogative) benchmark.

| Sup | Method | VidSTG (Interrogative) | | |
|---|---|---|---|---|
| | | m_vIoU | vIoU@0.3 | vIoU@0.5 |
| Full | TubeDETR [59] [CVPR2022] | 25.7 | 35.7 | 23.2 |
| | CSDVL [20] [CVPR2023] | 28.5 | 39.9 | 26.2 |
| | CG-STVG [15] [CVPR2024] | 29.0 | 40.5 | 27.5 |
| Weak | WINNER [29] [CVPR2023] | 10.2 | 12.0 | 5.5 |
| | VEM [21] [ECCV2024] | 13.3 | 16.7 | 7.7 |
| | CoSPaL [25] [ICLR2025] | 13.5 | 16.4 | 10.2 |
| ZS | ReCLIP [50] [ACL2022] | 8.4 | 8.0 | 2.3 |
| | E3M [3] [ECCV2024] | 10.6 | 12.2 | 5.4 |
| | Ours LLaVA-Next-Video-7B | 14.6 | 23.1 | 9.4 |
| | Ours Qwen2-VL-7B | 13.0 | 19.5 | 7.8 |
| | Ours ShareGPT4Video-8B | 13.1 | 21.0 | 9.1 |
| | Ours LLaVA-OneVision-7B | 13.5 | 21.4 | 8.2 |

Table 6: Comparison of different methods on RefCOCO series datasets.

| Sup | Method | RefCOCO | | | RefCOCO+ | | | RefCOCOg | |
|---|---|---|---|---|---|---|---|---|---|
| | | val | testA | testB | val | testA | testB | val | test |
| Full | MDETR [22] [CVPR2021] | 86.8 | 89.6 | 81.4 | 79.5 | 84.1 | 70.6 | 81.6 | 80.9 |
| | SeqTR [77] [ECCV2022] | 87.0 | 90.2 | 83.6 | 78.7 | 84.5 | 71.9 | 82.7 | 83.4 |
| | UNINEXT [32] [ACMMM2023] | 92.6 | 94.3 | 91.5 | 85.2 | 89.6 | 79.8 | 88.7 | 89.4 |
| | Shikra-7B [6] [arXiv] | 87.0 | 90.6 | 80.2 | 81.6 | 87.4 | 72.1 | 82.3 | 82.2 |
| | Ferret-7B [65] [ICLR2024] | 87.5 | 91.4 | 82.5 | 80.8 | 87.4 | 73.1 | 83.9 | 84.8 |
| Zero | ReCLIP [50] [ACL2022] | 45.8 | 46.1 | 47.1 | 47.9 | 50.1 | 45.1 | 59.3 | 59.0 |
| | ZS_REC [16] [CVPR2024] | 49.4 | 47.8 | 51.7 | 48.9 | 50.0 | 46.9 | 61.0 | 60.0 |
| | GroundVLP [46] [AAAI2024] | 65.0 | 73.5 | 55.0 | 68.8 | 78.1 | 57.3 | 74.7 | 75.0 |
| | Ours LLaVA-OneVision-7B | $68.7_{75.4\%}$ | $77.2_{80.8\%}$ | $62.0_{72.9\%}$ | $67.1_{74.4\%}$ | $76.7_{80.2\%}$ | $57.5_{67.7\%}$ | $72.3_{81.3\%}$ | $73.7_{83.2\%}$ |
| | Oracle | 91.1 | 95.6 | 85.0 | 91.2 | 95.6 | 84.9 | 88.9 | 88.6 |

## 8.2 Temporal Consistency Analysis

In our work, we obtain the attributes and actions related sub-queries. Specially, the decomposed attribute sub-query, which provides static state description, should be temporally consistent for spatial grounding. Interestingly, there exists evident inconsistency when introducing temporal augmentation in current MLLMs. We develop a temporal inconsistency metric by introducing reversing the order of input frames. In Fig. 15, we show the relation between spatial grounding accuracy and temporal

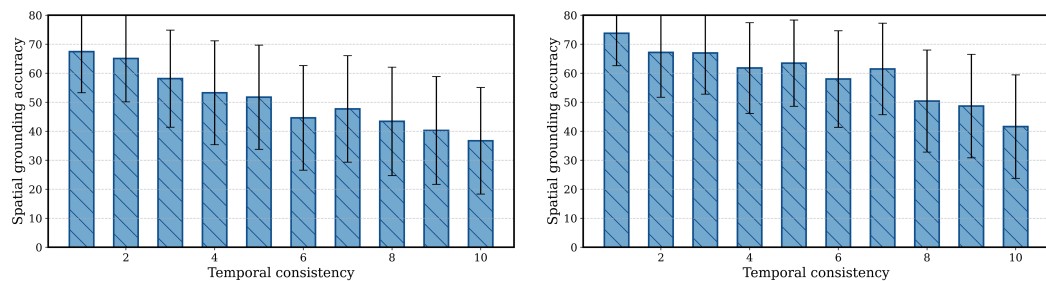

Figure 15: Spatial grounding accuracy of different groups of samples on the HC-STVGv1 dataset. These groups are ranked by descending temporal consistency.

Table 7: Iteration times $N_{ep}$ and learning rate lr ablation.

| $N_{ep}$ | lr | m_vIoU | vIoU@0.3 | vIoU@0.5 |
|---|---|---|---|---|
| 0 | 0 | 15.2 | 25.1 | 8.5 |
| 1 | 4 | 19.9 | 32.6 | 11.6 |
| **1** | **8** | **20.4** | **33.6** | **12.4** |
| 1 | 16 | 20.2 | 32.3 | 12.0 |
| 2 | 4 | 20.5 | 34.0 | 12.1 |
| 2 | 8 | 20.3 | 32.6 | 12.1 |
| 2 | 16 | 20.3 | 33.2 | 12.2 |

consistency based on `LLaVA-Next-Video-7B` model (left of Fig. 15) and `LLaVA-OneVision-7B` model (right of Fig. 15). We can find that although the `LLaVA-OneVision-7B` model achieves better localization performance, there is still the pronounced temporal inconsistency caused by temporal augmentation. Our findings provide guidance and insight for subsequent research in video MLLMs.

## 8.3 Effect of Optimization Times $N_{ep}$ and Learning Rate lr

The test-time tuning strategy DSTH is iterated $N_{ep}$ times for optimization with learning rate lr. Here we analyse effect of the hyper-parameters. As shown in Tab. 7, better results can be achieved as the optimization progresses and the our optimization strategy is relatively robust to these hyper-parameters. In particular, when setting $N_{ep} = 1$ and lr $= 8.0$, optimal performance is achieved in general.

## 8.4 Inference Efficiency

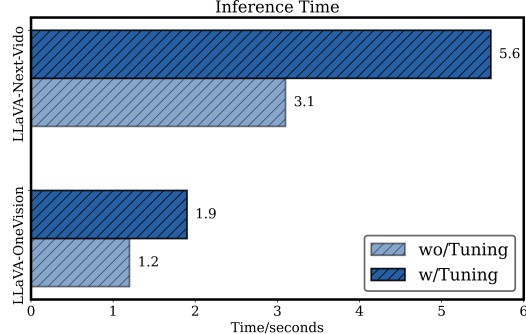

Figure 16: Comparison of inference time before/after introducing test-time tuning.

In Fig. 16, we show the inference efficiency of our proposed zero-shot framework. We test the inference speed on all video samples from HC-STVGv1 dataset on a single A100 GPU and then compute the average value. Specifically, before introducing the test-time tuning strategy DSTH for inference, our framework costs about 3.1 seconds for each video based on `LLaVA-Next-Video` model. After adopting the test-time tuning strategy, the cost is still acceptable though with some additional resource consumption. We believe that it would be more efficient to apply the resource-friendly inference schemes in the future.

### 8.5    Qualitative Spatio-Temporal Video Grounding Visualization

In Fig. 17, we present some video grounding examples for qualitative analysis. Here, we compare the results before introducing the test-time tuning (denoted with yellow boxes) with the results (denoted with green boxes) after introducing the optimization. The ground truth boxes are denoted with red boxes. By highlighting the attribute/action cues of the target by test-time tuning, our method can direct the MLLMs toward reliable visual context and improve spatial/temporal localization. We also give some failure cases (*e.g.*, the case (d)). Despite optimization during testing, the current model still pinpoints the wrong object instance. We attribute the less efficient comprehension to the poor visual conditions and suboptimal spatial inference in current MLLMs.

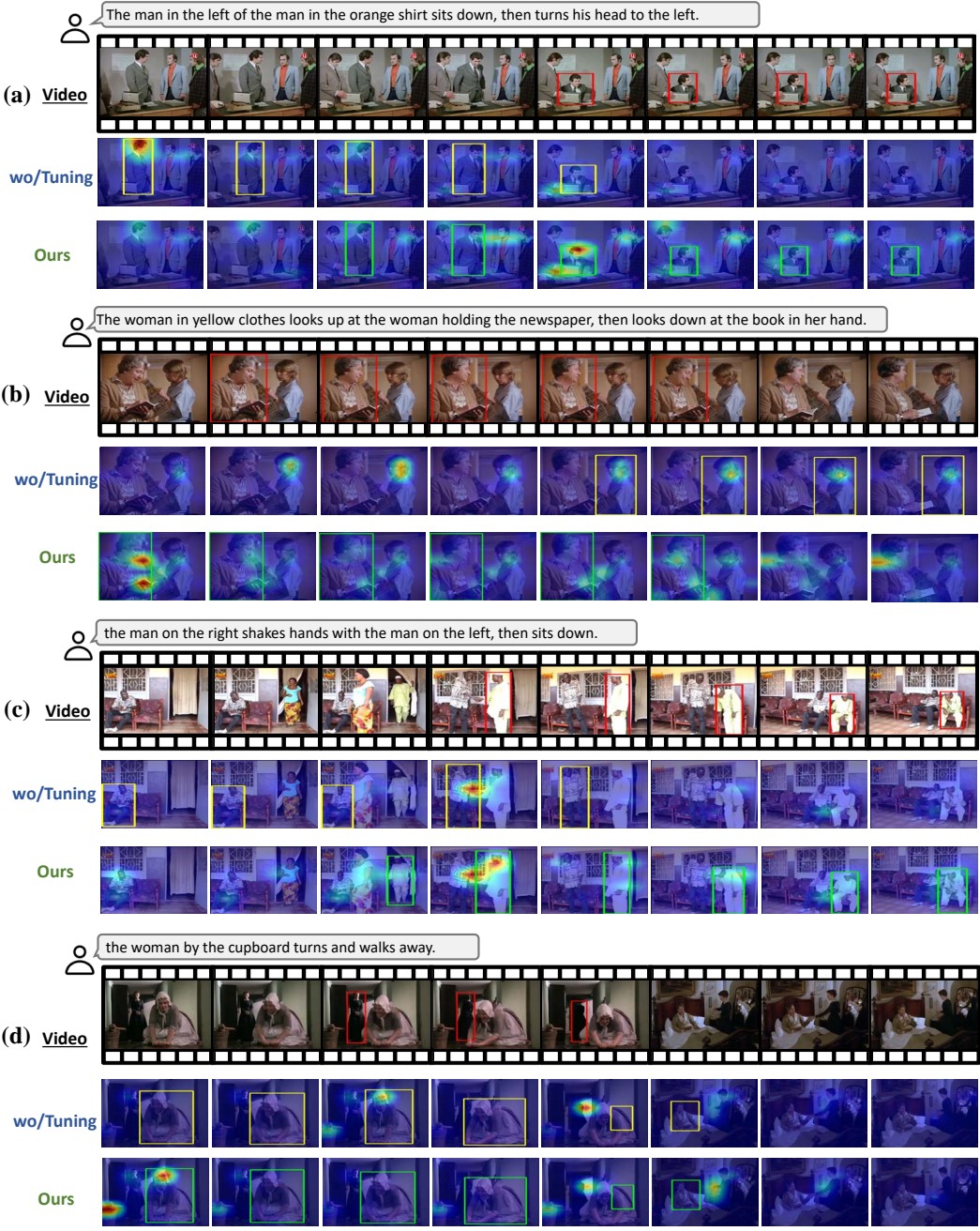

Figure 17: Comparison of attention maps on the HC-STVG dataset.

# 9    Limitation and Future work

Our work is simple yet effective, and also provides insights for related fields (*e.g.*, hallucination detection and attention-guided MLLMs pruning). Our work does have limitations. For example, based on MLLMs our framework can only receive a limited number of video frames as visual input due to the limit of computing resource. Besides, our framework needs to obtain attention for spatial and temporal inference, which is not compatible with the flash-attention mechanism adopted in current MLLMs. In the future, we will consider introducing a more efficient solution for the comprehension of long videos by incorporating token pruning and key frame selection techniques.

# 10    Broader Impacts

While we do not foresee our method causing any direct negative societal impact, it may potentially be leveraged by malicious parties to create applications that could misuse the grounding capabilities for unethical or illegal purposes. We urge the readers to limit the usage of this work to legal use cases.

