# OpenReview forum: "Unleashing the Potential of Multimodal LLMs for Zero-Shot Spatio-Temporal Video Grounding"
_NeurIPS.cc/2025/Conference — NeurIPS 2025 poster_

### Official Review · Reviewer_bp1T · 2025-06-30

**Clarity:** 3
**Significance:** 3
**Originality:** 3
**Rating:** 5
**Confidence:** 3

**Summary:**

The paper is about zero-shot STVG of multimodal LLMs. The authors first observe that special tokens subsequent to the text prompts used in several MLLMs have grounding abilities. The authors introduces training-free framework for STVG leveraging such special tokens with high visual activations. The framework begins with detecting object track proposals, followed by selecting tokens with highest attention value for locating target objects and getting object track score computed using max attention value within each track. The framework is further improved by decomposed spatio-temporal highlights strategy via interrogative subquery which highlights attributes and actions cues in language query.

**Questions:**

From the text in the checklist, it's not clear when and whether the authors will make their code available.
In the recent time, Qwen2.5-VL is widely used in place of Qwen2-VL. If the authors don't have enough time for new experiments; experiments on subsets can be considered to identify the trend.

**Ethical Concerns:**

["NO or VERY MINOR ethics concerns only"]

**Final Justification:**

I read the authors' responses to my and other reviewers' questions. I am satisfied with the responses.

**Limitations:**

The authors acknowledged that long-context scenario isn't addressed in this paper.

**Quality:**

3

**Strengths And Weaknesses:**

The method is simple and training-free. Query decomposition via interrogative sub-query and logit-guided re-attention for Q&A is intuitive and effective as shown in the results. The results on three benchmark datasets are strong, specially for the zero-shot and weakly supervised learning setup.

As a general note about these benchmarks: the video length is 20 seconds for HC-STVG Dataset and average video length is ~35 seconds for VidOR dataset on which the VidSTG dataset has been made off. Spatio-temporal video grounding as an application makes sense only if the videos are longer. Although this isn't a limitation for the proposed paper, but the benchmarks and the fields in general.

I see the core strength of the paper is in revealing the grounding capabilities of the special tokens. Training large models are expensive and there is a value in extracting meaningful information from pre-trained models.

---

> ### Author Rebuttal · Authors · 2025-07-30
>
> **Sincerely appreciate your recognition of our work! To the weaknesses and some questions, our response is as follows:**
>
> > **W1**: Spatio-temporal video grounding as an application makes sense only if the videos are longer. Although this isn't a limitation for the proposed paper, but the benchmarks and the fields in general.
>
> **[Ans]**: As you mentioned, long video comprehension remains a significant challenge with the limited context of LLM and computing resource.
>
> Here we also give some engineering optimization proposals. For example, we can try loading models by lower-quantization method (e.g., 4-bit). We can also optimize the inference process starting from the input. When performing temporal grounding, we can draw inspiration from previous works (e.g., **[A]** FastV and **[B]** VisionZip) to optimize less important visual tokens from a spatial perspective. Similarly, for spatial localization, we can selectively process a few key frames (e.g., **[C]** AKS) instead of inputting all video frames. These measures can effectively reduce inference latency in practical applications and enhance usability in long video understanding.
>
> We believe that integrating current research efforts to reduce resource consumption for video inference is necessary, and we will leave it for future works. Please let us know if you have any further questions or concerns.
>
> > **Q1**: From the text in the checklist, it's not clear when and whether the authors will make their code available.
>
> **[Ans]**: The original innovations in our work also have great significance and inspiration for other related fields (visual question answering, hallucination suppression, and explainability in MLLMs). To promote community research, we consider releasing the codebase once the review process is done.
>
> > **Q2**: In the recent time, Qwen2.5-VL is widely used in place of Qwen2-VL. If the authors don't have enough time for new experiments; experiments on subsets can be considered to identify the trend.
>
> We apologize for any inconvenience this may have caused. Our initial code development was based on version 4.48.0 of the Transformers library, which did not yet incorporate `Qwen2.5-VL`. After upgrading to version 4.50.0, we conducted experiments with `Qwen2.5-VL` on the `HC-STVG-v2` dataset. Considering that the `Qwen2.5` series includes models of various sizes—2B, 7B, 32B, and 72B—due to computational resource constraints (a single A100 GPU), we validated the effectiveness of our method on the smaller model sizes (2B and 7B). The results are shown in the following table:
>
> |                        | Vision | Language | N_hidden_layers | N_attention_heads | N_kv_heads | vlou@0.3 $\uparrow$ |  m_vlou $\uparrow$  |  m_tlou $\uparrow$  | gt_vlou $\uparrow$  |
> | ---------------------- | :---------: | :-----------: | :-------------: | :---------------: | :--------: | :------: | :------: | :------: | :------: |
> | Qwen2.5-VL-3B+baseline |     VIT     |  Qwen2.5-3B   |       36        |        16         |     2      |   28.5   |   17.3   |   37.1   |   39.1   |
> | Qwen2.5-VL-3B+Ours     |     VIT     |  Qwen2.5-3B   |       36        |        16         |     2      | **41.7** | **25.4** | **45.9** | **49.3** |
> | Qwen2.5-VL-7B+baseline |     VIT     |  Qwen2.5-7B   |       28        |        28         |     4      |   39.7   |   23.6   |   44.1   |   47.2   |
> | Qwen2.5-VL-7B+Ours     |     VIT     |  Qwen2.5-7B   |       28        |        28         |     4      | **45.4** | **27.7** | **46.9** | **51.6** |
>
> In the table, besides reporting grounding metrics (e.g., `vIoU@0.3`, `m_vIoU`), we also provide the configurations (e.g., `N_hidden_layers`) of different VL models.
>
> In addition, we introduce two additional metrics: `m_tIoU` and `gt_vIoU`. The `m_tIoU` metric only focuses on temporal grounding performance while ignoring spatial grounding, whereas the `gt_vIoU` metric focuses on spatial grounding performance while ignoring temporal grounding.
>
> From the results, we observe that our proposed method generalizes well to recent new MLLMs, demonstrating both the generality and scalability of our method. Moreover, we find that the model size significantly impacts the inherent grounding ability of special tokens. We also provide experiments on the Intern-VL3 series models in our response to `W3&Q3` of `Reviewer1-8wFh` for further verification of generality.
>
>
> Hope that our response can well addresses your concerns. Please let us know if you have any new questions or concerns.
>
>
> > [A] An Image is Worth 1/2 Tokens After Layer 2: Plug-and-Play Inference Acceleration for Large Vision-Language Models, ECCV2024
> >
> > [B] VisionZip: Longer is Better but Not Necessary in Vision Language Models, CVPR2025
> >
> > [C] AKS: Adaptive Keyframe Sampling for Long Video Understanding, CVPR2025

---

> > ### Comment · Reviewer_bp1T · 2025-08-06
> > **Authors' response**
> >
> > I thank the authors for providing the answers to my questions, specifically about the trend with recent MLLMs. They have shown the results for Qwen-2.5-VL 3B and 7B models, and also provided results for Intern-VL3 different sized models (to other reviewer's questions). I appreciate their empirical analysis of model size and their inherent grounding capabilities. It's no surprise that the improvements over smaller model size would be larger than their larger counterparts.

---

> > > ### Author Response · Authors · 2025-08-07
> > >
> > > Thank you very much for your positive feedback on the additional results analysis and your recognition of our work. We will incorporate the convincing results and analysis into the final version based on your thoughtful comments and release our code after the review process is complete.

---

### Official Review · Reviewer_LSgz · 2025-07-02

**Clarity:** 3
**Significance:** 3
**Originality:** 3
**Rating:** 4
**Confidence:** 4

**Summary:**

This paper proposes a zero-shot STVG framework based on MLLMs, which localizes target objects in videos according to text queries without requiring frame-level annotations. The framework enhances grounding capabilities through the introduction of DSTH and TAS strategies, and demonstrates strong performance across three benchmark datasets.

**Questions:**

(1) While the authors report the inference efficiency of their zero-shot framework, a broader comparison with other zero-shot, weakly-supervised, and fully-supervised methods would be valuable. How does the proposed method perform in terms of inference speed and computational complexity relative to these approaches?
(2) While the method achieves strong zero-shot performance on STVG, there is still a notable gap compared to fully-supervised approaches. Given that MLLMs themselves demonstrate strong performance in image spatial grounding and video temporal grounding, could the authors clarify why such a performance gap still exists?
(3) While the authors introduce a new method leveraging MLLMs for STVG, a natural baseline would be to directly use an MLLM to first perform temporal grounding, followed by spatial grounding on each frame. Could the authors clarify whether this straightforward approach would achieve comparable or even better performance than the proposed method? A comparison or analysis of this alternative would strengthen the paper.

**Ethical Concerns:**

["NO or VERY MINOR ethics concerns only"]

**Final Justification:**

The author's response has addressed most of my concerns, and the additional analysis effectively demonstrates the validity of their approach. Therefore, I am inclined to raise the initial score. I appreciate the author's efforts during the rebuttal process.

**Limitations:**

yes

**Paper Formatting Concerns:**

No.

**Quality:**

3

**Strengths And Weaknesses:**

Strengths
(1) The paper demonstrates that MLLMs’ ability to dynamically assign special tokens for visual grounding can be effectively used for precise zero-shot STVG without additional annotations.
(2) The method enhances performance across various MLLMs, demonstrating generalization capabilities.
(3) The ablation experiments fully demonstrate the effectiveness of the proposed test-time tuning strategy DSTH.
(4) The writing is clear and the illustrations are explicit.

Weaknesses
(1) In comparison with other methods, the reliance on MLLMs introduces higher computational complexity to this approach, which could potentially impact inference speed and efficiency.
(2) Since STVG concerns both temporal and spatial grounding, it is recommended to include a temporal grounding evaluation metric, such as m_tIoU, in addition to the metric m_vIoU. This approach, which has been adopted by recent methods like CG-STVG[1], CoSPaL[2] and TA-STVG[3], would provide a more comprehensive assessment.
[1] Context-guided spatio-temporal video grounding CVPR'24
[2] Contextual self-paced learning for weakly supervised spatio-temporal video grounding ICLR'25
[3] Knowing Your Target: Target-Aware Transformer Makes Better Spatio-Temporal Video Grounding ICLR'25

(3) The paper introduces a novel approach that leverages special markers in MLLMs for precise target localization. However, to comprehensively assess the model's performance, it is essential to provide a thorough analysis of failure cases. Currently, only a single failure case is presented in the supplementary materials, and there is a lack of in-depth analysis.

---

> ### Author Rebuttal · Authors · 2025-07-30
>
> **Sincerely thanks for insightful comments.   Your valuable reviews help  revise the manuscript. To the weaknesses and some questions, our response is as follows:**
>
> > **W1&Q1**: the reliance on MLLMs introduces higher computational complexity to this approach, which could potentially impact inference speed and efficiency.
>
> **[Ans]**: We report the inference time and compute metrics in the table below. Since there are few open-source works under weakly supervised and zero-shot settings, we compare our method with recent fully supervised works.
>
> Here we list the results on VitSTG dataset in the supplementary file of CG-STVG for reference. Also, we reproduce results by our computing device (one A100) on HC-STVG-v2 dataset. We used the **FlopCounterMode** method to measure the computational requirements when loading the model using 8-bit quantization.
>
> Our method does not fall significantly behind current methods in terms of inference efficiency. Instead, previous works require a significant amount of computational resources during training. To reduce inference latency and enhance practical feasibility, we have also provided some engineering optimization proposals in the section `Proposals for More Efficient Inference` as our reply to `W2 & Q2` of `Reviewer1-8wFh`.
> Please let us know if you have any further questions or concerns.
>
>    | Methods                             | Training | compute (Gflops) | Infer time (seconds) | throughput (Gflops/s) |
>    | :-- | :-: | :--: | :--: | :--: |
>    | CG-STVG                             | 32 A100  |      ~3000       |         0.6          |         ~5000         |
>    | TubeDETR                            | 16 V100  |      ~1500       |         0.4          |         ~3800         |
>    | STCAT                               | 32 A100  |      ~2900       |         0.5          |         ~5800         |
>    | CG-STVG (`reproduction`)            | 32 A100  |      ~1800       |         0.8          |         ~2200         |
>    | Qwen2-VL-7B (`reproduction`)        |   None   |      ~7400       |         2.8          |         ~2600         |
>    | LLaVa-OneVision-7B (`reproduction`) |   None   |      ~4000       |         1.9          |         ~2000         |
>
>
>
> > **W2**: Since STVG concerns both temporal and spatial grounding, it is recommended to include a temporal grounding evaluation metric, such as m_tIoU, in addition to the metric m_vIoU
>
> **[Ans]**: Thanks for careful suggestions!  In the table below, we have integrated the `m_tIoU` metric (**last column**) into the results comparison. Compared to the weakly-supervised SOTA method, our method achieves comparable performance on the temporal grounding evaluation metric `m_tIoU`. However, we notice that although our temporal grounding performance is slightly lower than the previous method **CoSPaL**, while our overall performance is significantly higher than **CoSPaL**. It maybe because that  MLLMs have weak differentiation between attention to different frames.  This indicates that there is still considerable room for improvement in temporal localization. We will add comparisons and discussions with these works to the final version.
>
> | Methods      | Supervision | m_vlou $\uparrow$ | vlou@0.3 $\uparrow$ | vlou@0.5 $\uparrow$ | m_tlou $\uparrow$ |
> | --- | :---: | :--: | -- | :---: | :--: |
> | CSDVL (CVPR2023)              |    Full     |  38.7  | 65.5     |   33.8   |  **58.1**  |
> | CG-STVG (CVPR2024)            |    Full     |  39.5  | 64.5     |   36.3   |  **60.0**  |
> | TA-STVG (ICLR2025)            |    Full     |  40.2  | 65.8     |   36.7   |  **60.4**  |
> | CoSPaL (ICLR2025)             |    Weak     |  22.2  | 31.4     |   18.9   |  **48.6**  |
> | **Ours (Qwen2-VL-7B)**        |    Zero     |  25.6  | 40.5     |   17.1   |  **45.2**  |
> | **Ours (LlaVa-OneVision-7B)** |    Zero     |  27.7  | 47.7     |   19.5   |  **46.0**  |
>
>
>
> > **W3**: To comprehensively assess the model's performance, it is essential to provide a thorough analysis of failure cases.
>
> **[Ans]**: Thanks for your careful comments! Due to the limitations of the rebuttal format, we regret that more failure cases cannot be provided immediately. Through our observations, in addition to the suboptimal tracking tubelet, the attention on the background token may negatively impacts grounding performance. Beisides, suboptimal spatial comprehension also cause unfaithful attendance to distractors as shown in Figure17(d) of supplement.  We will provide a more thorough analysis of failure cases in the final version.
>
>
>
> > **Q2**:  Given that MLLMs themselves demonstrate strong performance in image spatial grounding and video temporal grounding, could the authors clarify why such a performance gap still exists?
>
> **[Ans]**:
>
> - Though current MLLMs (e.g., Qwen2-VL) can achieve outstanding grounding performance by intorducing grounding datasets during instruction tuning, there is an obvious mechanistic difference between our attention-based, training-free solution and these fully-supervised methods. Our solution leverages the emergent grounding ability of special tokens, while these grounding models explicitly output the coordinates tokens corresponding to bounding boxes.
>
> - Even for MLLMs (e.g., LLaVA, DeepSeek-VL) without fine-tuning for target detection/grounding, they still exhibit emergent grounding abilities, as shown in Figure 2. This indicates that the grounding ability of special tokens may originate from cross-modal comprehension and alignment rather than fine-tuning for grounding tasks. As a result, the grounding ability of special tokens benefits little from the grounding-oriented fine-tuning process.
>
> - Here, we further analyze the bottleneck of grounding performance in our method. In the table below, we show the `m_tIoU`, which measures the performance of temporal grounding, and the `gt_vIoU`, which measures the performance of spatial grounding. We observe that there is a certain gap between our method and fully-supervised methods in temporal localization, which is an important factor affecting the overall performance. Additionally, we provide the upper bound of spatial grounding by selecting the candidate tubelet that best matches the ground truth in spatial grounding. The upper bound of spatial grounding is **65.8**, which suggests that we can improve performance by using better detectors and trackers for candidate tubelet generation.
>
>   | Methods       | Supervision | m_vlou $\uparrow$ | vlou@0.3 $\uparrow$ | vlou@0.5 $\uparrow$ | m_tlou $\uparrow$ | gt_vlou $\uparrow$ |
>   | -- | :--: | :--: | :--: | :--: | :--: | :---: |
>   | CSDVL (CVPR2023)                   |    Full     |  38.7  |   65.5   |   33.8   |  **58.1**  |   --    |
>   | CG-STVG (CVPR2024)                 |    Full     |  39.5  |   64.5   |   36.3   |  **60.0**  |   --    |
>   | TA-STVG (ICLR2025)                 |    Full     |  40.2  |   65.8   |   36.7   |  **60.4**  |   --    |
>   | CoSPaL (ICLR2025)                  |    Weak     |  22.2  |   31.4   |   18.9   |  **48.6**  |   --    |
>   | **Ours (Qwen2-VL-7B)**             |    Zero     |  25.6  |   40.5   |   17.1   |  **45.2**  |  50.2   |
>   | **Ours (LlaVa-OneVision-7B)**      |    Zero     |  27.7  |   47.7   |   19.5   |  **46.0**  |  53.1   |
>   | *Upper-Bound of Spatial Grounding* |             |        |          |          |            | *65.8*  |
>
>
>
> > **Q3**: A natural baseline would be to directly use an MLLM to first perform temporal grounding, followed by spatial grounding on each frame. Could the authors clarify whether this straightforward approach would achieve comparable or even better performance than the proposed method?
>
> **[Ans]**: Following your comments, we first perform temporal grounding, followed by spatial grounding on each frame. The results on HC-STVG-v2 by Qwen-2.5-VL (3B and 7B) models are given in this table.
>
> |         | Vision  | Language | N_hidden_layers | N_attention_heads | N_kv_heads | vlou@0.3 $\uparrow$ |  m_vlou $\uparrow$  |  m_tlou $\uparrow$ | gt_vlou $\uparrow$ |
> | -- | :----: | :---: | :---: | :--: | :--: | :--: | :---: | :---: | :---: |
> | Qwen2.5-VL-3B      |     VIT     |  Qwen2.5-3B   |       36        |        16         |     2      |   13.3   |   14.4   |   39.4   |   36.2   |
> | Qwen2.5-VL-3B+Ours |     VIT     |  Qwen2.5-3B   |       36        |        16         |     2      | **41.7** | **25.4** | **45.9** | **49.3** |
> | Qwen2.5-VL-7B      |     VIT     |  Qwen2.5-7B   |       28        |        28         |     4      |   17.8   |   15.2   |   35.7   |   41.3   |
> | Qwen2.5-VL-7B+Ours |     VIT     |  Qwen2.5-7B   |       28        |        28         |     4      | **45.4** | **27.7** | **46.9** | **51.6** |
>
> The **first** and **third** lines denote the results from token prediction by `Qwen2.5-VL` models. The **second** and **fourth** lines denote the performance of our method. We find that the temporal grounding (`m_tIoU`) and spatial grounding performance (`gt_vIoU`) are inferior to ours. We attribute this to the following points:
>
> - As we predict a bounding box for each frame individually, the objects localized in consecutive video frames often change, which significantly impacts the performance of spatial grounding.
> - Additionally, the VL models are sensitive to input prompts. Different prompts may lead to significantly different outputs. In the below, we show our prompts for temporal and spatial grounding. As a general-purpose model, Qwen-VL requires further fine-tuning to achieve superior performance on specific scenarios and tasks.
> ```
>  temporal  prompt : "Give the query: '{}', when does the described content occur in the video? please give the starting time (seconds) and ending time (seconds) in the json format."
>  spatial  prompt :  "In this image there is a person described by query '{}', please give his/her coordinates in the json format."
> ```
>
>
> Hope that our response can well addresses your concerns. Please let us know if you have any new questions or concerns.

---

> > ### Comment · Reviewer_LSgz · 2025-08-05
> >
> > The author's response has addressed most of my concerns, and the additional analysis effectively demonstrates the validity of their approach. Therefore, I am inclined to raise the initial score. I appreciate the author's efforts during the rebuttal process.

---

> > > ### Author Response · Authors · 2025-08-05
> > >
> > > Thanks very much for your positive feedback that we have addressed most of your concerns. We deeply appreciate that you are inclined to increase the initial score. We'll include the explanations and convincing results in our revision according to your careful comments, and release our code once the review process is done.

---

### Official Review · Reviewer_cWEf · 2025-07-02

**Clarity:** 2
**Significance:** 1
**Originality:** 1
**Rating:** 4
**Confidence:** 4

**Summary:**

This paper proposes a zero-shot spatio-temporal video grounding method. The authors observe that some special tokens within existing multi-modal LLMs (LLaVA-OneVision, Qwen2-VL, etc.) are superior for grounding, which they term _grounding tokens_. By leveraging this, the authors further propose methods to handle complex video queries: Decomposed Spatio-Temporal Highlighting (DSTH) and Logit-guided Re-Attention (LRA) to adapt grounding tokens for refining localization. The proposed method outperforms existing zero-shot methods using diverse off-the-shelf MLLMs on HCSTVG-v1&v2 and VidSTG datasets.

**Questions:**

- In Figure 6(b), why does using more frames (e.g., 11) lead to a drop in performance compared to using 7 frames? Could the authors elaborate on the possible reasons for this degradation?

**Ethical Concerns:**

["NO or VERY MINOR ethics concerns only"]

**Final Justification:**

The authors have addressed key concerns through a detailed rebuttal, providing both empirical evidence and analysis. I am raising my rating to borderline acceptance.

**Limitations:**

The authors briefly mention limitations in the conclusion section, but the discussion is not particularly constructive or detailed.

**Paper Formatting Concerns:**

No concerns.

**Quality:**

3

**Strengths And Weaknesses:**

**Strengths:**
- The paper is clearly written and easy to follow. Figures and graphs are helpful in illustrating the method and results.
- The study includes a comprehensive evaluation across a diverse set of open-source multimodal LLMs, including LLaVA-Next-Video, ShareGPT4Video, Qwen2-VL, and LLaVA-OneVision.

**Weaknesses:**
- The strongest existing baseline (E3M) does not use LLMs, and the performance gap when using LLaVA-Next-Video-7B appears marginal. Including analysis on model size and throughput would help contextualize this trade-off.
- The proposed method lacks clarity in some parts. The observation regarding the grounding behavior of special tokens is empirical, and the reliance on manually identifying grounding tokens per MLLM raises concerns about generalizability and scalability. When using larger models, such as 13B variants, does performance consistently improve? If not, could the authors provide insight into any observed saturation or trade-offs?
- According to the implementation details, the method depends on several additional modules, including Grounding DINO, SAM2, and GPT-4o. This multi-component design introduces potential efficiency and deployment concerns.

---

> ### Author Rebuttal · Authors · 2025-07-30
>
> **Sincerely thanks for insightful comments.   Your valuable reviews on our work  help  revise the manuscript for the final version. To the weaknesses and some questions, our response is as follows:**
>
> > **W1**: The strongest existing baseline (E3M) does not use LLMs, and the performance gap when using LLaVA-Next-Video-7B appears marginal. Including analysis on model size and throughput would help contextualize this trade-off.
>
> **[Ans]**:
>
> - **About the marginal performance gap.**
>   - The E3M adopts the **CLIP [A]** as the foundation model which is optimized toward discriminative pre-training objective and presents some advantages for target localization. Differently, our method is based on the LLMs which is toward generative pre-training objective. Unleashing the perception ability of generative models poses a greater challenge to the multimodal alignment of the foundation model.
>   - Despite the challenge, our work provides insightful observations about the grounding tokens of LLMs and presents a solution based on generative pre-training models. Compared to **E3M** based on discriminative pre-training models (e.g., **CLIP[A]**), our framework is simple and can well adapt to current multimodal LLMs and achieve comparable or even better grounding performance. When adopting the `LLaVA-OneVision-7B` model, our method outperforms 7.9% and 4.0% than the E3M method on the HC-STVG-v2 dataset.
>
>   - When adopting the less strong `LLaVA-Next-Video-7B` model, the grounding performance is inferior to the performance of LLaVA-OneVision-7B. But a substantial improvement has been achieved compared to the baseline using LLMs as shown in Table 2 of our manuscript, which well verifies the effectiveness of our method. In fact, the grounding performance may be further improved when integrating recent observations ([B],[C]) about discriminative layers into our framework. Overall, our work aims to reveal the inherent grounding ability of special tokens and propose a simple and effective grounding framework that can serve as a basis for further exploration in the future.
>
>
>
> - **Analysis on model size and throughput**
>
>   - Since there are currently few open-source works under weakly supervised and zero-shot settings, we compare our method with recent fully supervised works in terms of inference time and computational complexity in the following table.
>
>     | Methods                             | Training | compute (Gflops) | Infer time (seconds) | throughput (Gflops/s) |
>     | :-- | :-: | :-: | :-: | :--: |
>     | CG-STVG                             | 32 A100  | ~3000            | 0.6                  | ~5000                 |
>     | TubeDETR                            | 16 V100  | ~1500            | 0.4                  | ~3800                 |
>     | STCAT                               | 32 A100  | ~2900            | 0.5                  | ~5800                 |
>     | CG-STVG (`reproduction`)            | 32 A100  | ~1800            | 0.8                  | ~2200                 |
>     | Qwen2-VL-7B (`reproduction`)        | None     | ~7400            | 2.8                  | ~2600                 |
>     | LLaVa-OneVision-7B (`reproduction`) | None     | ~4000            | 1.9                  | ~2000                 |
>
>    - Here we list  the provided analysis results on VitSTG dataset in the supplementary file of CG-STVG for reference. Also, the lines with marker reproduction denote the reproduced results by our computing device (one A100) on HC-STVG-v2 dataset. We used the **FlopCounterMode** method to measure the computational requirements of **Qwen2-VL-7B** and **LLaVa-OneVision-7B** when loading the model using 8-bit quantization. Through comparison, it can be seen that our method does not fall significantly behind current methods in terms of inference latency and computational complexity. Instead, previous works require a significant amount of computational resources during training.
>
>    - **Proposals for more efficient inference**
>
>      Here we also give some engineering optimization proposals. For example, we can try loading models by lower-quantization method (e.g., 4-bit). We can also optimize the inference process starting from the input. When performing temporal grounding, we can draw inspiration from previous works (e.g., **[D]** FastV and **[E]** VisionZip) to optimize less important visual tokens from a spatial perspective. Similarly, for spatial localization, we can selectively process a few key frames (e.g., **[F]** AKS) instead of inputting all video frames. These measures can effectively reduce inference latency in practical applications and enhance usability in long video understanding. Overall, we think that the exploration of long video inference merits its own separate paper, and we leave it to future works.
>
>      Please let us know if you have further questions and concerns.
>
>
>
> > **W2**: The observation regarding the grounding behavior of special tokens is empirical, and the reliance on manually identifying grounding tokens per MLLM raises concerns about generalizability and scalability.
>
> **[Ans]**: Although the grounding behavior of special tokens is derived from empirical observations, it is well verified across different Multimodal LLMs, as shown in Figure 2 of the paper manuscript and the reply to `W3&Q3` of `Reviewer1-8wFh ` . Besides, in the reply to `W1&Q1`  of `Reviewer1-8wFh` we provide empirical justification.  The theoretical exploration of this phenomenon is also an interesting research topic, and we leave it to future works.
>
> Furthermore, our method does not require manually identifying grounding tokens for each MLLM. The special tokens equipped in each MLLM are fixed. As stated in `Line187–Line190`, we can effectively identify the optimal special token characterized by high visual activation based on our findings introduced in `Line 174`. This implementation is quite simple.
>
>
>
> > **W3**: the method depends on several additional modules, including Grounding DINO, SAM2, and GPT-4o. This multi-component design introduces potential efficiency and deployment concerns.
>
> **[Ans]**:  As in other zero-shot works (such as E3M and RedCircle), we utilize off-the-shelf detectors for tubelet generation. In addition, we also leverage GPT-4o for query decomposition. The overall pipeline design is simple and does not introduce excessive components.
>
> Furthermore, for these components, there are several lightweight alternatives available. For example, **[G]TinySAM** and **[H]EdgeTAM** introduce more lightweight designs based on the original SAM2 and can process video at almost real-time speed. For GPT-4o, there are also many lightweight alternatives, such as GPT‑4o Mini, GPT‑4.1 Mini, GPT‑4.1 Nano, and Gemini 2.5 Flash-Lite, which can reduce latency to the millisecond level. Thus, the introduced components have minimal impact during deployment in practical applications. We will release our code after acceptance.
>
>
>
> > **Q1**: In Figure 6(b), why does using more frames (e.g., 11) lead to a drop in performance compared to using 7 frames? Could the authors elaborate on the possible reasons for this degradation?
>
> **[Ans]**: We apologize for any confusion this may have caused.  Figure 6(b) shows the ablation study on the selected frames **K** during temporal prediction  instead of the input frames number. Since we need to simultaneously focus on both temporal and spatial localization, it means that for a target person we must localize the actions described in the text while ignoring other irrelevant temporal segments. When using more frames (e.g., 11) for temporal grounding prediction, the model tends to include more irrelevant temporal segments, leading to performance degradation. According to ablation results in Figure 6(b), we set the appropriate hyperparameter **K** .
>
> Hope that our response can well addresses your concerns. Please let us know if you have any new questions or concerns.
>
>
>
> > [A] Learning transferable visual models from natural language supervision, ICML2021
> >
> > [B] Unveiling Visual Perception in Language Models: An Attention Head Analysis Approach, CVPR2025
> >
> > [C] Layer by Layer: Uncovering Hidden Representations in Language Models, ICML2025
> >
> > [D] An Image is Worth 1/2 Tokens After Layer 2: Plug-and-Play Inference Acceleration for Large Vision-Language Models, ECCV2024
> >
> > [E] VisionZip: Longer is Better but Not Necessary in Vision Language Models, CVPR2025
> >
> > [F] AKS: Adaptive Keyframe Sampling for Long Video Understanding, CVPR2025
> >
> > [G] TinySAM: Pushing the Envelope for Efficient Segment Anything Model, AAAI2025
> >
> > [H] EdgeTAM: On-Device Track Anything Model, CVPR2025

---

> > ### Comment · Reviewer_cWEf · 2025-08-05
> > **post rebuttal comment**
> >
> > I appreciate the authors’ efforts to provide empirical evidence and concrete analysis. The clarifications provided have addressed my concerns.

---

> > > ### Author Response · Authors · 2025-08-05
> > >
> > > Thanks very much for your positive feedback that we have addressed your concerns.  We'll include the empirical evidence and concrete analysis in our final version according to your careful comments. We will release our code once the review process is done.

---

### Official Review · Reviewer_8wFh · 2025-07-13

**Clarity:** 3
**Significance:** 3
**Originality:** 4
**Rating:** 4
**Confidence:** 3

**Summary:**

This paper introduces novel decomposed spatio-temporal highlighting (DSTH) and TAS for robust zero-shot spatiotemporal video grounding with multimodal LLMs.
The DSTH is composed of action and attribute sub-queries, followed by the logit-guided re-attention (LRA) to highlight those cues.
The temporal-augmented assembling (TAS) strategy is also proposed to perform frame-level reversion operation.
It demonstrates the effectiveness on HCSTVG-v1, HCSTVG-v2, and VidSTG benchmarks.

**Questions:**

Q1 (for W1):
The LRA module contrasts “yes” and “no” logits to guide spatial/temporal attention, but this seems rather heuristic. Have you considered or evaluated alternative learning signals (e.g., entropy minimization, ranking loss)? Could you provide any theoretical or empirical justification for why this particular loss formulation leads to better grounding?

Q2 (for W2):
Test-time optimization (e.g., LRA) and the use of temporal-augmented inputs (TAS) likely increase inference time and memory usage. Could you report throughput (e.g., GFLOPS) comparison with other zero-shot or weakly-supervised STVG methods? Have you evaluated the practicality of the proposed method for real-time or long-video settings?

Q3 (for W3):
All experiments are conducted on 7–8B MLLMs. Have you tried smaller or larger MLLMs to validate the scalability and generality of your method? How sensitive is the performance to the model size?

**Ethical Concerns:**

["NO or VERY MINOR ethics concerns only"]

**Final Justification:**

The authors have addressed my concerns, so I keep my rating for acceptance.

**Limitations:**

yes

**Quality:**

3

**Strengths And Weaknesses:**

S1. This paper presents a novel zero-shot framework for spatio-temporal video grounding (STVG) that effectively exploits the grounding capabilities of special tokens in pre-trained MLLMs. The framework outperforms existing zero-shot methods.

S2. The proposed DSTH strategy decomposes a text query into attribute and action sub-queries and applies a novel LRA module for spatial and temporal prompt tuning. This is a practical solution to adapt frozen MLLMs during inference without finetuning.

S3. The paper offers compelling empirical evidence that MLLMs dynamically assign special tokens that exhibit strong grounding behavior. The analysis of attention maps across models adds a new perspective to the interpretability of MLLMs.

S4. The proposed method is evaluated on three STVG benchmarks and shows significant improvements over zero-shot, and even some weakly-supervised baselines. The method generalizes well across four different MLLMs, demonstrating robustness.

W1. While LRA effectively boosts grounding performance, the theoretical justification behind why contrasting logit responses (“yes” vs. “no”) leads to better grounding remains weak. The mechanism is intuitive but lacks formal analysis.

W2. The method involves prompt optimization at test time and uses additional modules (e.g., TAS). The inference cost (e.g., throughput or latency) comparisons against other zero-shot or weakly-supervised STVG methods are needed.
Also, the proposed method involves test-time optimization and uses dense visual tokens per frame. This may lead to high computational costs, making it less practical for long or real-time video processing.

W3. All experiments are conducted on 7–8B scale MLLMs. It would strengthen the claims of generality and scalability if the proposed method were validated across smaller (e.g., 3B) and larger (e.g., 13B, 30B) models as well.

W4. The visual pipeline in Figure 3 is informative, but the boundaries between key modules (e.g., DSTH, LRA, TAS) are not clearly demarcated. A more structured visualization would improve readability and make the method easier to understand.

---

> ### Author Rebuttal · Authors · 2025-07-30
>
> **We extend our sincere gratitude for your insightful reviews, which helps emphasize the contributions of our work.
> To the weaknesses and some questions, our response is as follows.**
>
> > **W1&Q1**: LRA module seems heuristic. Alternative learning signals and any theoretical or empirical justification is expected.
>
> **[Ans]**:
>
> - **Empirical justification**. Though contrasting logits of `yes` and `no` is heuristic, it is reasonable. We find that in some cases, the model often gives ambiguous or even wrong token/response predictions when given the question prompt. Here we show an example below.
>
>   ```Text
>   Query: Is there a woman by the cupboard turns in this video?
>
>   Answer (by llava-next-video): No, there is no woman visible in the video. The focus is on the group of men dressed in period clothing.
>   ```
>   However, the objects referred to by the text do indeed exist in the video. This indicates that the model cannot understand the video content well in certain cases. By optimizing the logit of the `yes` token toward positive prediction instead of the `no` token, we encourage the model to carefully attend to the target-related visual information and achieve better visual localization.
>
>   To validate our hypothesis, we measure the distribution of attention peaks before and after test-time optimization. Based on the evaluation on the HC-STVG-v2 dataset, we find that the average attention peak before test-time optimization is about **6.7** (without normalization), and after optimization, it increases to **23.9** (without normalization). **This change in attention values indicates that our proposed optimization strategy can encourage the model to perceive more visual modal information for generating visually-grounded responses**, thereby justifying our previous hypothesis.
>
> - **Alternative learning signals**. Here, we compare two approaches: `entropy minimization` (*i.e.,* increasing the predicted probability of the “yes” token across the entire vocabulary) and contrasting the logits of `yes` and `no` during test-time tuning. The results obtained by LLaVa-Next-Video-7B on the HC-STVG-v2 dataset are shown in the table below.
>
>   | Methods                                                      | Vision    | Language  | m_vlou $\uparrow$ | vlou@0.3 $\uparrow$ |  vlou@0.5 $\uparrow$ |
>   | -- | -- | -- | -- | -- | -- |
>   | LlaVA-Next-Video-7B + GTI     | clip-vit-large | vicuna/7B-V1.5 | 19.2   | 28.5     | 12.4     |
>   | LlaVA-Next-Video-7B + entropy minimization    | clip-vit-large | vicuna/7B-V1.5 | 19.5   | 28.5     | 12.5     |
>   | LlaVA-Next-Video-7B + contrasting the logits of `yes` and `no` | clip-vit-large | vicuna/7B-V1.5 | 22.1   | 32.9     | 14.0     |
>
>   The first line denotes the results of introducing grounding token identification (**GTI**). The second and third lines denote the test-time tuning results achieved by `entropy minimization` and contrasting the logits of `yes` and `no`, respectively. We find that `entropy minimization` does not bring noticeable improvement. We believe this is because `entropy minimization` may be achieved by reducing the token prediction of abundant irrelevant tokens in the vocabulary while ignoring the differentiation between the `yes` and the `no` token. As a result, it is less efficient than explicitly contrasting the logits of `yes` and `no` in promoting the model to attend to abundant visual information.
>
> > **W2&Q2**: Report throughput (e.g., Gflops) comparison with other STVG methods. Evaluated the practicality of the proposed method for real-time or long-video settings.
>
> **[Ans]**:
>
> - **Inference efficiency evaluation**.
>
>   Since there are few open-source works on weak supervision or zero-shot learning, we compared ours with recent fully supervised work in terms of inference efficiency in below table. We reproduce the inference time and computational complexity on the HC-STVG-v2 dataset following the setting (sample 20 frames with 336x336 resolution for each video) in manuscript  on our computing device (one A100-GPU).  We notice that CG-STVG provides an analysis of computation comparison on VidSTG in supplementary file. We utilize the information for reference.
>
>   | Methods       | Training | compute (Gflops) | Infer time (seconds) | throughput (Gflops/s) |
>   | :-- | :-: | :-: | :-: | :--: |
>   | CG-STVG        | 32 A100  | ~3000            | 0.6                  | ~5000                 |
>   | TubeDETR            | 16 V100  | ~1500            | 0.4                  | ~3800                 |
>   | STCAT      | 32 A100  | ~2900            | 0.5                  | ~5800                 |
>   | CG-STVG (`reproduction`)            | 32 A100  | ~1800            | 0.8                  | ~2200                 |
>   | Qwen2-VL-7B (`reproduction`)        | None     | ~7400            | 2.8                  | ~2600                 |
>   | LLaVa-OneVision-7B (`reproduction`) | None     | ~4000            | 1.9                  | ~2000                 |
>
>   Specifically, we used the **FlopCounterMode** method to measure the computational requirements of **Qwen2-VL-7B** and **LLaVa-OneVision-7B** when loading the model using 8-bit quantization. We mainly evaluate the required compute during the visual tokens preparation and language model inference. Through comparison, it can be seen that our method does not fall significantly behind current methods in terms of inference latency and computational complexity. Instead, previous works require a significant amount of computational resources during training.
>
> - **Proposals for more efficient inference**
>
>   In practice, in addition to using lower-quantization loading methods (e.g., 4-bit), we can also optimize the inference process starting from the input. For example, when performing temporal grounding, we can draw inspiration from previous works (e.g., [A] FastV and [B]VisionZip) to optimize less important visual tokens from a spatial perspective. Similarly, for spatial localization, we can selectively process a few key frames (e.g.,  [C] AKS) instead of inputting all video frames. These measures can effectively reduce inference latency in practical applications and enhance usability in long video understanding. Overall, we think that the exploration of long video inference merits its own separate paper, and we leave it to future works.
>
> > **W3&Q3**: Generality and scalability of proposed method can be validated across smaller and larger models as well.
>
> **[Ans]**:  We conduct the scalability verification on Intern-VL series models. With limited computational resources (one A-100), we evaluate our method on **InternVL3-2B**, **InternVL3-8B** and **InternVL3-14B** in the table below. Besides reporting the grounding metric (e.g., `vIoU@0.3`, `m_vIoU`), we also give the config (e.g, `N_hidden_layers`) of different VL models.
>
> - Generality of our method is well validated across different scales of models. As indicated in the main paper, we average the attention maps of all special tokens as the **baseline**. Benefiting from our grounding token identification and test-time optimization, we can obviously achieve better grounding performance not only in the smaller (4B) but also the larger models (14B).
> - Models with larger number of parameters often yield better results. The performance of small models (e.g, `InternVL3-2B`) is far inferior to the larger models. It is because fewer parameters in language models perform poorly in multimodal alignment. By further probing the attention maps, we find that the attention maps are often centered on tokens in background and edge regions. How to unlock the interpretability of small models' attention maps is an interesting research topic, and we leave it for future work. Besides, the performance of  `InternVL3-14B` declines slightly than the `InternVL3-8B`, we think it is because it has too many layers and attention maps with grounding ability are easily overwhelmed by a large number of attention maps with poor grounding ability.
>
> |           |     Vision     |   Language   | N_heads | N_layers | N_kv_heads | vIoU@0.3 $\uparrow$ |  m_vIoU $\uparrow$  |   t_iou $\uparrow$   | gt_viou $\uparrow$ |
> | :- | :-: | :-: | :-: | :-: | :-: | :-: | :-: | :-: | :-: |
> | InternVL3-2B+baseline   | InternViT-300M | Qwen2.5-0.5B |   12    |    28    |     2      |   25.3   |   16.0   |   35.9   |   38.4   |
> | InternVL3-2B+Ours       | InternViT-300M | Qwen2.5-0.5B |   12    |    28    |     2      | **27.7** | **17.8** | **38.3** | **40.0** |
> | InternVL2.5-4B+baseline | InternViT-300M |  Qwen2.5-3B  |   16    |    36    |     2      |   24.5   |   17.0   |   38.4   |   38.2   |
> | InternVL2.5-4B+Ours     | InternViT-300M |  Qwen2.5-3B  |   16    |    36    |     2      | **41.1** | **25.4** | **45.5** | **49.5** |
> | InternVL3-8B+baseline   | InternViT-300M |  Qwen2.5-7B  |   28    |    28    |     4      |   36.5   |   22.9   |   42.9   |   46.1   |
> | InternVL3-8B+Ours       | InternViT-300M |  Qwen2.5-7B  |   28    |    28    |     4      | **45.3** | **27.5** | **47.2** | **53.2** |
> | InternVL3-14B+baseline  | InternViT-300M | Qwen2.5-14B  |   40    |    48    |     8      |   33.2   |   21.0   |   41.2   |   44.8   |
> | InternVL3-14B+Ours      | InternViT-300M | Qwen2.5-14B  |   40    |    48    |     8      | **42.2** | **26.2** | **45.8** | **51.3** |
>
> > **Q4**: boundaries between key modules in Figure 3 can be more clearly demarcated.
>
> **[Ans]**: Thanks for your constructive comments on paper revision. We will revise the boundaries between key modules in Figure 3, and make it clearly demarcated in the final version.
>
> Hope that our response can well addresses your concerns. Please let us know if you have any other concerns.
>
> > [A] An Image is Worth 1/2 Tokens After Layer 2: Plug-and-Play Inference Acceleration for Large Vision-Language Models, ECCV2024
> >
> > [B] VisionZip: Longer is Better but Not Necessary in Vision Language Models, CVPR2025
> >
> > [C] AKS: Adaptive Keyframe Sampling for Long Video Understanding, CVPR2025

---

> > ### Comment · Reviewer_8wFh · 2025-08-07
> >
> > Thanks to the authors for considering my concerns. I have one quick question for W2&Q2, is the computation, especially for Qwen2 and LLaVA, measured over the entire process?

---

> > > ### Author Response · Authors · 2025-08-07
> > >
> > > Thanks very much for your positive feedback. Yes, we measured the computation across the entire process, including both visual token preparation and LLM inference. We observed that, with the greater optimization of language models, visual preparation can account for a non-negligible portion of the computational cost. As a result, precomputing visual features could be a viable option for improving efficiency. Please feel free to reach out if you have any further questions.

---

### Note · Authors · 2025-08-13

We extend our sincere gratitude to all the reviewers for insightful reviews, which help us emphasize the contributions of our work and revise the presentation. We also sincerely thank AC and SAC for the time and effort.

- **Contribution**

We are encouraged to hear that the reviewers found the work is well-motivated with good presentation and contributions as well as a comprehensive evaluation.  Here we list the **three core contributions**: (1) We reveal the superior grounding ability of special tokens characterized by the salient visual activation in MLLMs.  (2) We propose a novel zero-shot STVG framework, which includes innovative logit-guided test-time tuning and temporal-augmented assembling to further unleash the ability of video comprehension.
(3) We conduct extensive experiments on various MLLMs to demonstrate the generalization of our findings and the proposed method.

- **Rebuttal**

A concern among several reviewers is the inference latency of our method with the introduction of LLMs. We systematically compare our method and previous works, and verify that our method has comparable performance in terms of inference efficiency. Besides, we provide more experimental results to verify the generality and scalability of the proposed method.

We are pleased to hear that our careful rebuttal has addressed the reviewers' concerns. We will revise our manuscript and incorporate the new results in the final version based on reviewers'  suggestions, and release the code once the review process is done.

---

### Decision · Program_Chairs · 2025-09-17

**Decision:**

Accept (poster)

**Comment:**

The paper introduces a framework for zero-shot spatiotemporal video grounding using pre-trained multimodal large language models (MLLMs). The central contribution is the empirical observation that pre-trained MLLMs contain special grounding tokens that are particularly effective for localizing concepts expressed by text queries, enabling zero-shot grounding capability. Initial reviews were mixed, but following the rebuttal, all reviewers recommended acceptance. Despite minor concerns regarding some technical and experimental details, the reviewers agreed that the identification of grounding tokens is insightful, the proposed method is novel, and the empirical results are extensive. The AC concurs with the reviewers' post-rebuttal assessment and recommends acceptance. The authors are encouraged to incorporate the rebuttal materials into the camera-ready version.